# Towards Bridging the Gap Between Offline and Iterative Alignment via Preference Distillation

## Abstract

Direct preference optimization (DPO, Rafailov et al., 2024) is a promising offline approach for aligning large language models (LLMs) due to its simplicity, computational efficiency, and implicit modeling of human preferences. Interestingly, iterative extensions of DPO have achieved stronger performance on academic benchmarks, raising two key questions: (i) Why do iterative methods generally outperform offline ones? (ii) Can their advantages be incorporated into offline alignment? To answer the first question, our controlled experiments reveal that the *explicit preference model*, additionally introduced in the iterative procedure, is a key factor behind its superiority over offline methods. This insight leads us to answer the second question affirmatively and propose Distilled Preference Probability Policy Optimization (DP3O), an effective and efficient offline alignment algorithm. DP3O first learns an explicit preference model using a helper class of LLMs and then distills its knowledge into policy optimization. Theoretically, we show that explicit preference modeling admits better estimation error control than implicit formulations, and that DP3O achieves a tighter generalization bound than hard-label DPO through variance reduction. Empirically, we evaluate DP3O on a wide range of chat-based and downstream tasks and show that it outperforms state-of-the-art offline methods, achieves performance comparable to iterative DPO, and reduces training time by about 42%, demonstrating both its effectiveness and efficiency.

## 1 Introduction

Aligning large language models (LLMs) using human feedback data can greatly improve the abilities of pre-trained models to follow instructions (Stiennon et al., 2020; Agrawal et al., 2023; Rafailov et al., 2024). A prominent approach to tackle the problem of learning from human preference data is through reinforcement learning from human feedback (RLHF, Ziegler et al., 2019; Ouyang et al., 2022; Yuan et al., 2024a). While RLHF refines LLMs with impressive conversational and coding abilities, RLHF methods like proximal policy optimization (PPO, Schulman et al., 2017) often struggle with computational inefficiency and instability. In contrast, some recent offline alternatives, such as direct preference optimization (DPO, Rafailov et al., 2024) have emerged as a viable solution, because they are simpler to implement and require fewer computing resources by using the policy to implicitly model human preference.

Despite the practical success of DPO (MistralAI, 2023; Xu et al., 2024b), the standard offline version often falls short compared to its *iterative* variant. Typically, the iterative approach starts by training a preference model and then iteratively annotates data generated by the LLMs for further fine-tuning. Recent empirical studies find that despite the training complexities, the iterative algorithms like the iterative DPO (Xu et al., 2023) show superior performance in academic benchmarks (see e.g., Tajwar et al., 2024; Tang et al., 2024; Dong et al., 2024; Xiong et al., 2024). This observation raises two interesting questions:

*(i) Why do iterative methods generally outperform their offline counterparts?*
*(ii) Can we incorporate the strengths of the iterative procedure in purely offline settings?*

In this paper, we first conduct controlled experiments to address the first question and identify the explicit preference model, which is additionally used in the iterative procedure, as the key factor contributing to the gap between the offline and iterative methods. Motivated by this observation, we propose a novel offline

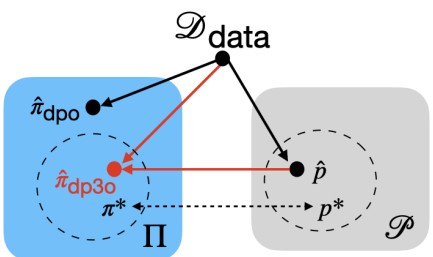

Figure 1: **Intuition behind our method**: Both the explicit preference model $\hat{p}$ and the implicit model $\hat{\pi}_{\text{dpo}}$ aim to approximate the ground-truth preference model $p^*$. For the implicit class, searching for $p^*$ is equivalent to searching for the optimal policy $\pi^*$. In practice, we observe that learning $p^*$ is easier within the explicit preference class $\mathcal{P}$ than within the implicit policy class $\Pi$. As a result, $\hat{\pi}_{\text{DP3O}}$ leverages the fitted explicit model as a form of distillation to improve policy alignment.

algorithm called Distilled Preference Probability Policy Optimization (DP3O) that synthesizes the strengths of both offline and iterative methods to optimize the alignment of LLMs with human preferences. The intuition of our method is illustrated in Fig. 1. DP3O consists of two steps: (i) learn an explicit preference model from the offline data using a helper class of LLMs, and (ii) distill the knowledge from the explicit model to the implicit one as soft labels.

The key **contributions** of our work can be summarized as follows.

• We design a series of controlled experiments and identify the critical role of the explicit preference model in explaining the gap between the offline and the iterative methods. Our empirical results address existing discrepancies in the literature, offering practical guidance for the application of RLHF methods.
• We propose an effective and efficient offline alignment method, DP3O, that incorporates a preference-distilled cross-entropy, which efficiently transfers knowledge from an explicit preference model to an implicit preference framework. Compared with iterative methods, our approach directly distills soft labels from the explicit preference model, reducing the computational cost associated with extensive response generation and annotation. Compared with the offline methods, DP3O achieves better performance by incorporating additional knowledge into policy learning.
• We provide a theoretical analysis showing that explicit preference modeling admits better estimation error control than implicit formulations, and that DP3O can generalize better than DPO when the explicit preference model is sufficiently accurate.
• Empirically, through extensive evaluations on controlled experiments and real-world chat-based evaluation tasks, including AlpacaEval 2, MT-Bench and also various downstream tasks in Open LLM Leaderboard, DP3O consistently outperforms existing state-of-the-art offline methods. These results establish DP3O as a highly effective framework for LLM alignment.

## 2 Related Works

**RLHF Algorithms**. RLHF plays a crucial role in aligning large language models (LLMs) with human preferences and has been successfully applied in state-of-the-art models such as ChatGPT (Achiam et al., 2023), Claude (Bai et al., 2022), and the Llama series (Touvron et al., 2023; Dubey et al., 2024). Proximal Policy Optimization (PPO) (Schulman et al., 2017) is one of the most prominent algorithms in the literature on large language model (LLM) alignment. However, it faces notable challenges due to its instability, inefficiency, and sensitivity to implementation specifics (Engstrom et al., 2020). Moreover, PPO often requires additional components, such as a reward model and a value network, which significantly increases the demand for GPU memory resources during training (Zheng et al., 2023b). Thus, researchers have attempted to design alternative approaches for LLM alignment to resolve the aforementioned issues. There are some works aiming to optimize the LLMs without extra components, including Rafailov et al. (2024); Azar et al. (2024); Liu et al. (2024); Ethayarajh et al. (2024). Among these methods, Direct Preference Optimization (DPO) (Rafailov et al., 2024) is the first proposed offline alignment approach, offering a compelling alternative to PPO with improved stability and competitive performance. Our work also relates to variants of DPO that utilize soft labels. CDPO (Mitchell, 2023) and RDPO (Chowdhury et al., 2024) address noisy preference labels by incorporating constant soft labels. Concurrently, Furuta et al. (2024) proposes a geometric averaging approach that weights samples based on the preference gap using reward model scores. In reward learning, Zhu et al. (2024) introduces iterative data smoothing, updating labels toward learned preferences to mitigate reward overoptimization. Additionally, Rashidinejad & Tian (2024) dynamically updates preference labels during training to create stationary labels, reducing gradients for unreliable samples. Some prior work also

distills external reward signals for offline preference optimization (Fisch et al., 2024; Chen et al., 2024; Ji et al., 2024; Sun et al., 2025b). For example, InfoNCA (Chen et al., 2024) uses soft-label distillation, but is formulated as contrastive learning over more than two responses with a softmax parameterization and assumes access to ground-truth rewards rather than binary preferences. Distilled DPO (d-DPO) (Fisch et al., 2024) directly regresses proxy rewards via MSE with an implicit policy. While d-DPO recognizes the benefits of explicit reward modeling, our controlled experiments further identify it as a substantial factor in the offline–iterative gap, and we show both empirically and theoretically that distilling preference *probabilities* offers advantages over hard labels. Moreover, DP3O interpolates between hard-label and soft-label from distillation, allowing a trade-off that mitigates performance loss from imperfect distillation. There are also some recent studies (Lin et al., 2024; Razin et al., 2026) that compare explicit and implicit modeling paradigms, analyzing their differences in terms of generalization.

**Offline vs. Iterative RLHF.** Current alignment methods can be broadly classified into two categories: offline and iterative. Offline methods usually refer to contrastive alignment methods, such as DPO and IPO, which utilize static offline preference data to fine-tune the model without generating new preference data. PPO is a notable example of a step-wise iterative approach that generates new data at each step of the training process. Recent research (Tran et al., 2023; Xiong et al., 2024; Rosset et al., 2024; Cen et al., 2024; Gao et al., 2024) has introduced iterative versions of offline methods that leverage an external model or human feedback to provide preference signals. Additionally, Yuan et al. (2024b); Wu et al. (2024) employ the training language model itself through LLM-as-a-Judge prompting to generate feedback for new data. Some studies (Zhang et al., 2024; Xiong et al., 2024; Xie et al., 2024) also explore hybrid approaches that combine offline/historical datasets with newly generated data for model updates. Empirically, most findings suggest that iterative variants significantly outperform traditional offline methods. To investigate the reasons behind this performance gap, Tang et al. (2024) proposed hypotheses from multiple perspectives, including data characteristics, optimization processes, loss functions, and scaling properties, and conducted empirical experiments to validate or challenge these hypotheses. Additionally, Tajwar et al. (2024) examined the conditions under which techniques like on-policy sampling and contrastive losses can enhance the performance of offline algorithms. Both studies overlook a critical factor: the quality of the reward model. In this work, we aim to explore this aspect to further illustrate the performance gap between offline and iterative methods.

## 3 Preliminary

### 3.1 Problem Formulation and Notations

We study learning from preference-based feedback to align LLMs. Define a generic language model as a policy $\pi : \mathcal{X} \to \Delta(\mathcal{Y})$, which maps a context (prompt) $x \in \mathcal{X}$ to an action (response) $y \in \mathcal{Y}$ based on the probability distribution $\pi(y \mid x)$. Here $\Delta(\cdot)$ is referred to as a class of all probability distributions over the corresponding space. We use $\Pi$ to denote the policy class and $\mathcal{P}$ for the explicit preference model class. Given a pre-trained LLM $\pi_{\mathrm{ref}}$ learned from supervised fine-tuning (SFT), we collect a preference-based dataset. Specifically, a prompt is first generated by a probability distribution $\rho$. Then, a pair of responses $(y_1, y_2)$ is sampled from $\pi_{\mathrm{ref}}$ independently conditioned on $x$. Lastly, a human annotator is asked to label the preference between $y_1$ and $y_2$. We denote $y_w$ as the preferred response and $y_l$ as the less preferred one among $(y_1, y_2)$, i.e., $y_w \succ y_l$ and $(y_w, y_l)$ is the preference order transformation of $(y_1, y_2)$. Assume that the preference order $(y_w, y_l)$ is generated based on the following preference distribution given $(x, y_1, y_2)$.

$$p^\star (y_1 \succ y_2 \mid x) = g(r^\star (x, y_1) - r^\star (x, y_2)), \tag{1}$$

where the preference function $g : \mathbb{R} \to [0, 1]$ is a monotone non-decreasing function satisfying $g(z) + g(-z) = 1$ and $r^\star : \mathcal{X} \times \mathcal{Y} \to \mathbb{R}$ is the latent true reward function in each human annotator's mind. Then our preference-based dataset consists of $n$ i.i.d. copies of $(x, y_w, y_l)$ denoted by $\mathcal{D}_{\mathrm{off}} = \{x^{(i)}, y_w^{(i)}, y_l^{(i)}\}$. Given $\mathcal{D}_{\mathrm{off}}$, the goal is to find a policy $\pi$ to maximize:

$$\mathbb{E}_{x \sim \rho, y \sim \pi(\cdot|x)} [r^\star(x, y)] - \beta D_{\mathrm{KL}} (\pi \| \pi_{\mathrm{ref}}(x)),$$

where $D_{\mathrm{KL}} (\pi \| \pi_{\mathrm{ref}}) := \mathbb{E}_{x \sim \rho, y \sim \pi(\cdot|x)} \log \frac{\pi(y|x)}{\pi_{\mathrm{ref}}(y|x)}$ is the reverse Kullback–Leibler (KL) divergence penalty, which encourages the learned policy to stay close to $\pi_{\mathrm{ref}}$. The objective is to maximize the true rewards while limiting deviation from the reference policy.

### 3.2 RLHF for Alignment

Since the reward function $r^\star$ is unknown, the RLHF approach is to first learn $r^\star$ in (1) based on $\mathcal{D}_{\mathrm{off}}$. For example, let $g(z) = 1/(1 + \exp(-z))$ and Model (1) becomes the Bradley-Terry (BT) (Bradley & Terry, 1952), which is arguably the most popular preference model. Then one can get an estimated reward function by minimizing the cross-entropy loss with respect to $\phi$:

$$-\mathbb{E}_{(x,y_w,y_l)\sim\mathcal{D}_{\mathrm{off}}} \left[\log \sigma \left(r_\phi \left(x, y_w\right) - r_\phi \left(x, y_l\right)\right)\right], \tag{2}$$

where $\phi$ is a parameter used to model $r^\star$. Once the resulting estimator $\widehat{\phi}$ is obtained, in the second step of the direct approach, one can learn the policy $\hat{\pi}$ by

$$\max_\theta \mathbb{E}_{x\sim\rho, y\sim\pi_\theta(\cdot|x)} \left[r_{\widehat{\phi}}(x, y)\right] - \beta D_{\mathrm{KL}} \left(\pi_\theta \| \pi_{\mathrm{ref}}\right). \tag{3}$$

Solving (3) typically involves iterative sampling and optimization. Algorithms like PPO and GRPO (Schulman et al., 2017; Shao et al., 2024) have shown promise in aligning LLMs with human preferences but face significant limitations. They require multiple components, such as value networks and reward models, which demand substantial GPU memory, introduce training instability, and increase computational complexity with numerous hyperparameters (Zheng et al., 2023b; Ivison et al., 2024).

### 3.3 Direct Preference Optimization and Its Iterative Variant

While RLHF has been shown to achieve remarkable empirical success, Rafailov et al. (2024) argued that RLHF, as a two-step approach, is a complex and often unstable procedure due to fitting a reward model in (2) and performing penalized policy learning in (3). Instead of using the estimate-reward-then-optimize approach, Rafailov et al. (2024) proposed DPO, which directly learns $\pi^*$ by utilizing the policy model for the reward model. In particular, their approach is motivated by the following observation (Peters & Schaal, 2007; Peng et al., 2019): the closed-form solution of (3) with a generic function $r$ for any $(x, y)$ is

$$\pi_\theta(y \mid x) = \frac{1}{Z(x)}\pi_{\mathrm{ref}}(y \mid x) \exp \left(\frac{1}{\beta}r(x, y)\right),$$

where $Z(x) = \sum_y \pi_{\mathrm{ref}}(y \mid x) \exp \left(r(x,y)/\beta\right)$ is called the partition function. Then equivalently $r$ can be represented via $\pi_\theta$:

$$r(x, y) = \beta \log \frac{\pi_\theta(y \mid x)}{\pi_{\mathrm{ref}} (y \mid x)} + \beta \log Z(x), \tag{4}$$

which we term implicit preference modeling. Thanks to the implicit preference modeling, together with equation (2), DPO method proposes learning $\pi^\star$ via minimizing $\mathcal{L}_{\mathrm{DPO}} \left(\pi_\theta; \pi_{\mathrm{ref}}\right)$:

$$-\mathbb{E}_{(x,y_w,y_l)\sim\mathcal{D}_{\mathrm{off}}} \left[\log \sigma \left(\beta \log \frac{\pi_\theta \left(y_w \mid x\right)}{\pi_{\mathrm{ref}} \left(y_w \mid x\right)} + \cancel{\beta \log Z(x)} \quad -\beta \log \frac{\pi_\theta \left(y_l \mid x\right)}{\pi_{\mathrm{ref}} \left(y_l \mid x\right)} - \cancel{\beta \log Z(x)}\right)\right], \tag{5}$$

with respect to $\theta$. Denote the estimated optimal policy from (5) as $\widehat{\pi}_{\mathrm{DPO}}$. We call this DPO policy as an **implicit preference model**.

**Iterative DPO**: Iterative DPO (e.g., Xu et al., 2023; Xiong et al., 2024) was introduced to improve DPO's empirical performance with three stages. In the first stage, offline pairwise data is used to fit an **explicit preference model**. In the second stage, assume at iteration $t$, given prompt set distribution $\rho_t$, for each prompt $x \sim \rho_t$, the current policy $\pi_t$ generates $K$ responses $\{y_k\}_{k=1}^K$, and the fitted preference model is used to identify the most and least preferred pairs based on the highest preference probability.

$$y_w, y_l = \underset{y_i, y_j \in \{y_k\}_{k=1}^K}{\arg\max} \quad p_{\widehat{\phi}}(y_i \succ y_j \mid x). \tag{6}$$

We can denote $y_w \sim \pi_t^{(K)}(\cdot \mid x)$ and $y_l \sim \pi_t^{(1)}(\cdot \mid x)$, where two policies are called best-of-n and worst-of-n policies respectively (Pace et al., 2024; Gui et al., 2024). Under BT parameterization, this is equivalent to

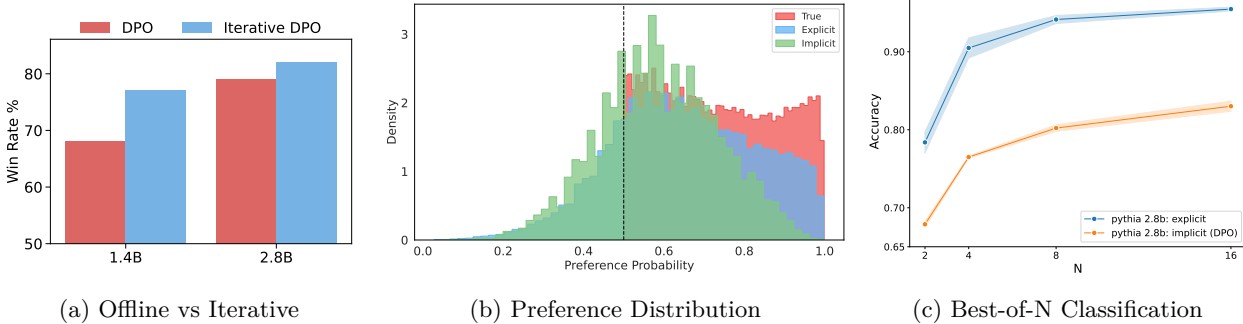

(a) Offline vs Iterative      (b) Preference Distribution      (c) Best-of-N Classification

Figure 2: (a) Performance comparison between offline and iterative DPO methods, measured by win rate compared with SFT. Iterative DPO consistently outperforms its offline variant. (b) The distribution of preference probabilities annotated by the implicit and explicit preference model. Explicit preference probability is closer to the true probability (c) Classification accuracy of preference models on generated samples shows that explicit models outperform the implicit model.

maximizing the reward difference. In the last stage, standard DPO is then applied to the sampled data to update the policy. The second and third stages can be repeated for $T$ iterations, with each iteration typically using a non-overlapping subset of prompts comprising $1/T$ of the total.

Iterative DPO is computationally expensive due to repeated response generation and annotation. Hence we propose DP3O, an effective and efficient offline algorithm that incorporates advantages of iterative methods.

## 4 DP3O: Preference Probability Distilled Policy Optimization

### 4.1 Explicit Preference Model Matters

Iterative DPO has been observed to outperform offline DPO in two distinct settings: in the purely offline case, where only a fixed dataset is available (Tang et al., 2024; Swamy et al., 2025), and in scenarios where the preference model is trained on a large corpus of external preference data, as opposed to a relatively smaller offline dataset (Tajwar et al., 2024; Xiong et al., 2024). In the latter case, the performance improvement is expected, as iterative methods have access to more extensive and diverse data than offline DPO. In contrast, in the purely offline setting, one would anticipate similar performance between two methods, given that both approaches rely on the same dataset. According to the data processing inequality (MacKay, 2003), transforming raw data through a learned preference model cannot increase its information content; therefore, additional sampling from the initial policy should not provide gains beyond the original dataset.

We aim to investigate the factors contributing to the performance gap between offline and iterative DPO methods when trained on the same initial dataset.

**Controlled Experimental Setup.** We conduct controlled experiments on the Anthropic HH-RLHF dataset (Bai et al., 2022). We first train a golden reward model $r^*$ based on Mistral-7B-Instruct-v0.2 (Jiang et al., 2023), and then use it to annotate and rerank pairwise preference data under the Bradley–Terry (BT) model. We also use this golden reward model for evaluation. The overall pipeline for data generation, training, and evaluation is illustrated in Appendix Fig. 5.

For the backbone models, we use the supervised fine-tuned Pythia series (Biderman et al., 2023), including Pythia-1.4B-SFT and Pythia-2.8B-SFT. We set the number of iterations to 3. For explicit preference modeling, we adopt the Bradley–Terry parameterization and initialize the preference model from the same SFT checkpoint as that used by both offline and iterative DPO. The explicit preference model contains an additional linear head for reward prediction. In contrast, offline DPO optimizes preferences only implicitly through the policy objective in Eq. (5), without introducing a separate preference model. To control confounders, we use the same optimizer, learning rate, and batch size across all experiments.

**A Performance Gap Exists.** We begin by comparing offline DPO and iterative DPO. As shown in Fig. 2a, the iterative variant consistently outperforms the offline ones when varying model sizes. This result suggests that iterative alignment benefits from access to additional preference information accumulated across iterations.

Prior work (Tajwar et al., 2024; Song et al., 2024; Xiong et al., 2024) has largely attributed the advantage of iterative methods to on-policy data generation, which allows the model to learn from its own responses and gradually correct its mistakes. While this explanation is likely part of the story, our results suggest that it is not the whole story. Another important factor is the use of *explicit* preference modeling.

To isolate this factor, we distinguish between two forms of preference modeling:

*Explicit preference model:* $p_\phi(y_1 \succ y_2 \mid x) = \sigma\left(r_\phi(x, y_1) - r_\phi(x, y_2)\right)$.

*Implicit preference model:* $p_\theta(y_1 \succ y_2 \mid x) = \sigma\left(\log \frac{\pi_\theta(y_1|x)}{\pi_{\text{ref}}(y_1|x)} - \log \frac{\pi_\theta(y_2|x)}{\pi_{\text{ref}}(y_2|x)}\right)$.

The key distinction is that the explicit model directly parameterizes logits through a separate reward function based on hidden representations, whereas the implicit model represents preferences through policy likelihood ratios. Since iterative methods maintain an explicit preference model while offline DPO does not, this distinction provides a plausible explanation for part of the observed performance gap.

**Explicit Preference Models Better Capture Preferences.** We next compare the two types of preference models directly. Specifically, we evaluate them from two perspectives: (1) how well they fit preference distribution, and (2) how accurately they judge preferences among generated responses.

For preference distribution fitting, we use each trained model to estimate preference probabilities on the validation data. As shown in Fig. 2b, the explicit model matches the ground-truth preference probabilities more accurately than the implicit model, even though both are trained under the same optimization settings. This result indicates that the explicit model achieves better preference distribution fitting.

To evaluate preference judgment, we test whether each model can correctly identify the most preferred and least preferred responses among multiple candidates generated from test prompts. For each prompt, we sample between 2 and 16 candidate responses, ask each model to select the best and worst responses, and compute the corresponding selection accuracy. Higher accuracy indicates stronger preference judgment ability. As shown in Fig. 2c, the explicit model consistently outperforms the implicit model across all settings, demonstrating stronger preference discrimination ability during generation.

**Motivation.** This raises an important question: can we leverage the information captured by the explicit preference model to improve the training of the implicit model used in offline alignment without online data generation? If we can distill knowledge from the explicit model, we may be able to narrow the performance gap between offline and iterative methods. This insight forms the core motivation for our DP3O algorithm.

## 4.2 Algorithm

Firstly, based on (4), we parameterize the implicit preference model $p_\theta(y_w \succ y_l \mid x)$ as

$$p_\theta(y_w \succ y_l \mid x) = \sigma\left(\beta \log \frac{\pi_\theta(y_w \mid x)}{\pi_{\text{ref}}(y_w \mid x)} - \beta \log \frac{\pi_\theta(y_l \mid x)}{\pi_{\text{ref}}(y_l \mid x)}\right).$$

The DPO objective in (5) can be viewed as a cross-entropy loss over preference labels. Specifically, let $H(p, q) := -p \log q - (1 - p) \log(1 - q)$ denote the binary cross-entropy between two Bernoulli probabilities $p$ and $q$, and let $H(p) := -p \log p - (1 - p) \log(1 - p)$ denote the binary entropy. Then,

$$\mathcal{L}_{\text{DPO}} = \mathbb{E}[H(p^*, p_\theta)] = \mathbb{E}[D_{\text{KL}}(p^* \| p_\theta)] + H(p^*), \tag{7}$$

where $H(p^*)$ is constant with respect to $\theta$. Therefore, minimizing (7) is equivalent to minimizing $D_{\text{KL}}(p^* \| p_\theta)$.

Motivated by this, we introduce a distillation objective defined as the cross-entropy between $p_{\hat\phi}$ and $p_\theta$:

$$\mathcal{L}_{\text{distill}} := \mathbb{E}_{(x, y_1, y_2) \sim \mathcal{D}_{\text{off}}} H(p_{\hat\phi}, p_\theta),$$

---

**Algorithm 1** DP3O

---

**Require:** $\mathcal{D}_{\text{off}} = \{x, y_w, y_l\}_{i=1}^n$
1: Train explicit preference model $p_{\hat{\phi}}$ on $\mathcal{D}_{\text{off}}$ by minimizing $-\mathbb{E}_{(x,y_w,y_l)\sim\mathcal{D}_{\text{off}}}\left[\log p_{\hat{\phi}}(y_w \succ y_l \mid x)\right]$
2: For each $\{x, y_w, y_l\} \in \mathcal{D}_{\text{off}}$ , use $\hat{p}_\phi$ to score the preference probability.
3: Optimize $\mathcal{L}_{\text{DP3O}}\left(\pi_\theta; \pi_{\text{ref}}, p_{\hat{\phi}}\right)$ in (8) to obtain $\pi_{\hat{\theta}}$
4: **return** $\hat{\pi} = \pi_{\hat{\theta}}$

---

which is equivalent to minimizing $D_{\text{KL}}(p_{\hat{\phi}}\|p_\theta)$ up to the constant entropy term $H(p_{\hat{\phi}})$.

We then combine the distillation objective with the original DPO objective:

$$\begin{aligned}\mathcal{L}_{\text{DP3O}}(\pi_\theta; \pi_{\text{ref}}, p_{\hat{\phi}}) &= \alpha\mathcal{L}_{\text{DPO}}(\pi_\theta; \pi_{\text{ref}}) + (1-\alpha)\mathcal{L}_{\text{distill}}(\pi_\theta; \pi_{\text{ref}}, p_{\hat{\phi}})\\ &= -\mathbb{E}_{(x,y_1,y_2)\sim\mathcal{D}_{\text{off}}}\Big[((1-\alpha)p_{\hat{\phi}}^+ + \alpha)\log p_\theta^+ + (1-\alpha)p_{\hat{\phi}}^- \log p_\theta^-\Big],\end{aligned} \quad (8)$$

which corresponds to interpolating between hard preference labels and soft targets induced by $p_{\hat{\phi}}$. $\alpha \in [0, 1]$ is an imitation parameter balancing the influence of the soft labels provided by the explicit preference model and hard labels, and $p_\bullet^+ = p_\bullet(y_w \succ y_l \mid x)$, $p_\bullet^- = p_\bullet(y_w \prec y_l \mid x)$ ($\bullet$ denotes a placeholder for the model's parameterization.). This objective can be viewed as a preference-weighted optimization that leverages the adjusted distilled preference probabilities $(1-\alpha)p_{\hat{\phi}}^+ + \alpha$ and $(1-\alpha)p_{\hat{\phi}}^-$ as soft targets. Accordingly, we refer to this approach as Distilled Preference Probability Policy Optimization (DP3O). Alternatively, it can be interpreted as a form of dynamic label smoothing, where the smoothing factor is given by the adjusted preference probability.

In practice, DP3O is implemented as a two-stage approach. First, the explicit preference model $p_\phi$ is fitted using $\mathcal{D}_{\text{off}}$. Next, the predicted preference probability $p_{\hat{\phi}}$ is used as a plug-in to solve (8) and obtain the optimized $\theta$, as outlined in Algorithm 1.

Compared with iterative methods, our method doesn't have the heavy computational burden and hence is highly efficient. Our approach can be viewed as a middle ground between DPO and iterative methods, combining the efficiency and stability of offline training with the guidance of explicit preference modeling from iterative/online RLHF. As a result, it offers a balanced trade-off between performance and computational cost, making it a practical and scalable solution for preference optimization.

Unlike related soft-label methods (Ji et al., 2024; Sun et al., 2025b), our approach preserves the original DPO objective and introduces an imitation term to bridge the objectives. These methods can be seen as special cases of ours, and experiments underscore the value of retaining the DPO objective.

We also conduct a gradient analysis on $\mathcal{L}_{\text{DP3O}}$ to further understand our method. Note that

$$\begin{aligned}&\nabla_\theta\mathcal{L}_{\text{DP3O}}\left(\pi_\theta; \pi_{\text{ref}}, p_{\hat{\phi}}\right)\\ =&\nabla_\theta - \mathbb{E}\Big[((1-\alpha)p_{\hat{\phi}}^+ + \alpha)\log p_\theta^+ + (1-\alpha)p_{\hat{\phi}}^- \log p_\theta^-\Big]\\ =&-\beta\mathbb{E}\Big[\underbrace{\Big(\alpha + (1-\alpha)p_{\hat{\phi}}^+ - \sigma(r_\theta(x,y_w) - r_\theta(x,y_l))\Big)}_{\text{difference of pref prob. between exp \& implicit}} \cdot \Big[\underbrace{\nabla_\theta\log\pi(y_w \mid x)}_{\text{increase likelihood of } y_w} - \underbrace{\nabla_\theta\log\pi(y_l \mid x)}_{\text{decrease likelihood of } y_l}\Big]\Big],\end{aligned}$$

where $r_\theta(x,y) = \beta\log\pi_\theta(y \mid x)/\pi_{\text{ref}}(y \mid x)$ is the reward implicitly defined by the language model $\pi_\theta$ and reference model $\pi_{\text{ref}}$. Intuitively, the gradient of $\mathcal{L}_{\text{DP3O}}$ leverages the difference between the adjusted explicit preference probability $\alpha + (1-\alpha)p_{\hat{\phi}}^+$ and the implicit preference probability $\sigma(r_\theta(x,y_w) - r_\theta(x,y_l))$ to adjust response likelihoods. A positive difference, indicating the implicit model is underconfident, increases the likelihood of $y_w$ and decreases that of $y_l$, improving confidence. Conversely, a negative difference, indicating overconfidence, decreases the likelihood of $y_w$ and increases that of $y_l$, mitigating overconfidence.

### 4.3 Theoretical Analysis

#### 4.3.1 Estimation Error of the Preference Model

We have empirically observed that the explicit preference model is easier to fit than the implicit preference model. To better understand this difference, we next study the estimation error of the preference model.

**Theorem 4.1.** *Let $p^\star(y_1 \succ y_2 \mid x)$ denote the true pairwise preference probability for $x \in \mathcal{X}$ and $y_1, y_2 \in \mathcal{Y}$. Let $\rho \in \Delta(\mathcal{X})$ be a prompt distribution and $\pi(\cdot \mid x) \in \Delta(\mathcal{Y})$ a response distribution. For each $i \in [n]$, sample $x^{(i)} \sim \rho$ and $y_1^{(i)}, y_2^{(i)} \overset{\text{i.i.d.}}{\sim} \pi(\cdot \mid x^{(i)})$, then draw a preference according to $p^\star(y_1^{(i)} \succ y_2^{(i)} \mid x^{(i)})$. Let $(y_w^{(i)}, y_l^{(i)})$ be the induced winner-loser ordering, and define $\mathcal{D} = \{(x^{(i)}, y_w^{(i)}, y_l^{(i)})\}_{i=1}^n$.*

*Let $\mathcal{F}$ be a finite class of functions $p : \mathcal{X} \times \mathcal{Y} \times \mathcal{Y} \to [0,1]$ such that $p^\star \in \mathcal{F}$, and assume that $0 < p(y_1 \succ y_2 \mid x) < 1$ for all $p \in \mathcal{F}$ and all $(x, y_1, y_2) \in \mathcal{X} \times \mathcal{Y} \times \mathcal{Y}$. Let*

$$\widehat{p} := \arg \max_{p \in \mathcal{F}} \sum_{(x, y_w, y_l) \in \mathcal{D}} \log p(y_w \succ y_l \mid x).$$

*Then for any $\delta \in (0,1)$, with probability at least $1 - \delta$,*

$$\mathbb{E}_{x \sim \rho, \, y_1, y_2 \sim \pi(\cdot \mid x)} \left[ D_{\mathrm{H, Ber}}^2 \big( \hat{p}(y_1 \succ y_2 \mid x), \, p^\star(y_1 \succ y_2 \mid x) \big) \right] \leq \frac{2 \log(|\mathcal{F}|/\delta)}{n},$$

*where $D_{\mathrm{H, Ber}}^2(a, b) := D_{\mathrm{H}}^2(\mathrm{Ber}(a), \mathrm{Ber}(b))$ and $D_{\mathrm{H}}^2(P, Q) := \int \left( \sqrt{p(z)} - \sqrt{q(z)} \right)^2 dz.$*

The proofs are deferred to Appendix B. The theorem bounds the estimation error of the learned preference model by $\frac{2 \log(|\mathcal{F}|/\delta)}{n}$, where $|\mathcal{F}|$ is the size of the hypothesis class. Since real preference models are usually large neural networks with high-dimensional parameter spaces rather than a literal finite class, Theorem 4.1 mainly captures the scaling of estimation error with the effective hypothesis-class size, rather than providing a sharp finite-sample guarantee.

In our controlled setting, the explicit and implicit preference models share the same backbone and training data, but induce different hypothesis classes due to their parameterizations. Let $\mathcal{P}$ and $\Pi$ denote the hypothesis classes induced by the explicit and implicit parameterizations, respectively.

In particular, the implicit class $\Pi$ can be richer than the explicit class $\mathcal{P}$. Even with the upstream backbone shared across the two families, the implicit head reuses the LM head $\mathbb{R}^d \to \mathbb{R}^V$ with $V \times d$ parameters and yields a reward realized as a sum of $T$ per-token log-ratios over a $V$-dim vocabulary, while the explicit BT head is a single $\mathbb{R}^d \to \mathbb{R}$ linear layer with only $d$ parameters that collapses every response to a scalar. This structural gap is further confirmed by a empirical randomization-test experiment inspired by Zhang et al. (2017): with an identical backbone and same optimization, the implicit family can fully memorize randomly flipped preference labels, while the explicit family stays at the random-guessing and cannot memorize them. We defer the parameter analysis, the randomization-test experimental results to Appendix A.4.

Therefore, under the same model scale, the theorem suggests that the implicit parameterization may admit a looser estimation bound, and hence potentially slower statistical convergence, than the explicit one. We note that this comparison is specific to the matched-scale setting and does not necessarily generalize to cases where the explicit and implicit models have different scales.

#### 4.3.2 Understanding the Benefits of Preference Distillation via Risk Decomposition

The previous theorem suggests that the explicit preference model $\hat{p}$ can better approximate the true preference distribution $p^*$. We now explain how this benefits the alignment policy $\pi$ when $\pi$ is viewed as an implicit preference model $p_\pi$. Specifically, we study how distillation from $\hat{p}$ affects the risk of $p_\pi$ with respect to $p^*$ via a risk decomposition. In the analysis below, we treat $\hat{p}$ as fixed and do not model the dependence induced by training both models on the same data.

**Definition 4.2.** (Population risk and Empirical risk). We define the population risk and empirical risk of DPO and DP3O, respectively, as:

$$R(\pi) := -\mathbb{E}_{\mathcal{D}}\Big[p^*\left(x, y_l, y_w\right)^{\top} \log p_{\pi}\left(x, y_l, y_w\right)\Big],$$

$$\hat{R}_{\mathrm{DPO}}(\pi; \mathcal{D}) := -\frac{1}{n} \sum_{i=1}^{n} \log p_{\pi}\left(y_w^{(i)} \succ y_l^{(i)} \mid x^{(i)}\right),$$

$$\hat{R}_{\mathrm{DP3O}}(\pi; \mathcal{D}, \hat{p}) := -\frac{1}{n} \sum_{i=1}^{n} \Big[\hat{p}_{\alpha}\left(x^{(i)}, y_l^{(i)}, y_w^{(i)}\right)^{\top} \log p_{\pi}\left(x^{(i)}, y_l^{(i)}, y_w^{(i)}\right)\Big].$$

where $p(x, y_l, y_w) := [\, p(y_w \succ y_l \mid x),\, p(y_w \prec y_l \mid x)\,]$ for the corresponding binary preference vector, and define its smoothed version as $p_{\alpha}(x, y_l, y_w) := [\,(1-\alpha)p(y_w \succ y_l \mid x) + \alpha,\, (1-\alpha)p(y_w \prec y_l \mid x)\,]$.

The population risk measures the expected alignment between the implicit policy and the true preference distribution, while the empirical risk is its finite-sample approximation. We then show that using true preference probabilities in DP3O yields a lower-variance objective than DPO.

**Lemma 4.3.** *For any policy $\pi \in \Pi$ and $\mathcal{D} \sim \mathbb{P}_{\mathcal{D}}$ :*

$$\mathbb{V}_{\mathcal{D}}\hat{R}_{DP3O}(\pi; \mathcal{D}, p^*) \leq \mathbb{V}_{\mathcal{D}}\hat{R}_{DPO}(\pi; \mathcal{D}),$$

*where $\mathbb{V}$ denotes variance.*

The result follows from a variance decomposition and Jensen's inequality, and proofs are deferred to Appendix B. The two variances become equal only in degenerate cases, such as when the preference probability is constant or takes values only in $\{0, 1\}$. Since preferences in RLHF are typically stochastic, true preference probabilities provide more information than hard labels alone. Therefore, DP3O with access to true preferences generally has lower objective variance than DPO.

Since we don't know the ground truth human preference, we can only utilize the fitted preference model $\hat{p}$ as the proxy, and we show the generalization bound when using the proxy:

**Theorem 4.4.** *For any policy $\pi \in \Pi$ and $\hat{p} \in \mathcal{P}$. Then for any $\delta \in (0, 1)$, with probability at least $1 - \delta$ over $\mathcal{D} \sim \mathbb{P}_{\mathcal{D}}$ :*

$$R(\pi) \leq \hat{R}_{DP3O}(\pi; \mathcal{D}, \hat{p}) + \mathcal{O}\left(\sqrt{\hat{\mathbb{V}}_{\mathcal{D}}(\pi, \hat{p}) \cdot \frac{\log \frac{|\Pi|}{\delta}}{n}} + \frac{\log |\Pi|}{n}\right) + \mathcal{O}\left((1-\alpha)\,\mathbb{E}_{\mathcal{D}}\,\|\hat{p} - p^*\|_2\right).$$

The second term reflects the empirical variance of DP3O objective. As shown in Lemma 4.3, using a true soft target can reduce this variance, and this effect may generalize when employing an explicit preference model. The final term acts as a penalty, capturing the discrepancy between the explicit and true preference models. Note that this bound is shared by DPO and DP3O: DPO corresponds to the special case $\alpha = 1$, while DP3O is the general $\alpha \in [0, 1)$ case. Theorem 4.4 characterizes a bias-variance tradeoff: DP3O reduces the variance term relative to DPO (Lemma 4.3) at the cost of a teacher-discrepancy term that vanishes for DPO. Whether the variance reduction outweighs the introduced bias depends on the teacher quality and the data distribution; the gain is therefore problem-dependent rather than universal. In our experiments, the explicit preference model is sufficiently accurate; the observed improvement of DP3O over DPO is consistent with this favorable bias-variance regime.

## 5 Experiments

### 5.1 Controlled Experiments

We further train and evaluate DP3O in controlled experiments and compare DP3O against several baselines, including offline methods such as DPO (Rafailov et al., 2024), its label-smoothing variants CDPO (Mitchell,

Table 1: Controlled experiments: comparison of DP3O and baseline methods in terms of win rate, reward gain $\Delta r$ relative to SFT, and training time. **Bold** indicates the best results among offline methods.

| Type | Method | Win Rate % ($\uparrow$) | | | $\Delta r$ ($\uparrow$) | Training Time (h) ($\downarrow$) |
|------|--------|------|--------|---------|--------|---------|
| | | Base | Chosen | Average | | |
| Offline | DPO | 78.62 | 47.90 | 63.26 | 1.22 | 0.92 |
| | CDPO | 71.18 | 38.16 | 54.67 | 0.87 | 0.91 |
| | RDPO | 73.52 | 43.07 | 58.30 | 1.05 | 0.91 |
| | d-DPO | 79.51 | 49.66 | 64.59 | 1.33 | 1.54 |
| | DP3O | **82.65** | **53.24** | **67.95** | **1.39** | 1.55 |
| Iterative | Iterative DPO it-1 | 53.04 | 29.45 | 41.24 | 0.16 | - |
| | Iterative DPO it-2 | 78.01 | 45.66 | 61.84 | 1.20 | - |
| | Iterative DPO it-3 | 84.30 | 61.09 | 72.69 | 1.71 | 2.67 |

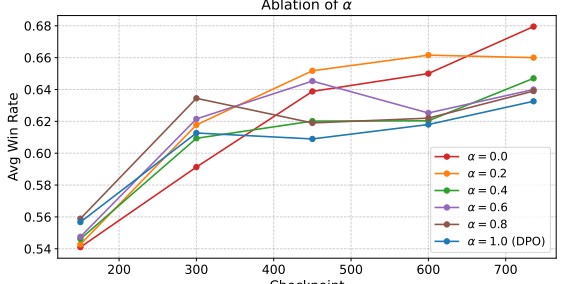 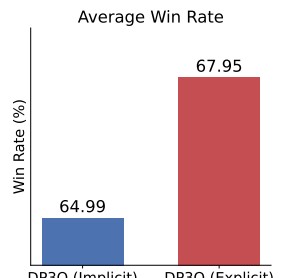 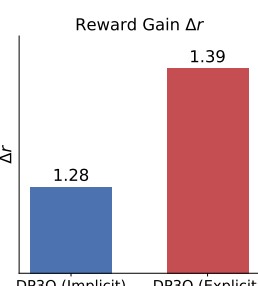

Figure 3: Ablation on $\alpha$ and teacher model type. **Left**: Average win rate of DP3O under different values of $\alpha$, using the explicit preference model as the default teacher. **Right**: Comparison of DP3O with two types of teacher models, evaluated by average win rate and reward gain $\Delta r$.

2023) and RDPO (Chowdhury et al., 2024), and the reward-distillation method d-DPO (Fisch et al., 2024), as well as iterative DPO. All methods are instantiated with Pythia 2.8B. We evaluate all methods using two metrics derived from the true reward function $r^\star$: win rate and reward gap $\Delta r$. Win rate measures the fraction of pairwise comparisons won by each method, while $\Delta r$ quantifies the difference in true rewards between the outputs of $\pi_\theta$ and $\pi_{\text{sft}}$. Additional details are provided in Appendix A.1.

**DP3O consistently outperforms offline baselines, matches iterative DPO, and is more computationally efficient.** As shown in Table 1, DP3O consistently outperforms all offline baselines across the controlled experiments. This supports our hypothesis that informative soft targets provide a stronger training signal than either hard labels alone or constant soft labels, as used in label-smoothing-based methods. In particular, the gains over CDPO and RDPO suggest that the benefit comes not merely from smoothing the preference labels, but from distilling a more informative preference distribution. Moreover, DP3O also outperforms d-DPO, indicating that directly distilling preference probabilities can be more effective than distilling reward signals. We further compare DPO and DP3O in terms of preference prediction accuracy in Appendix Fig. 6. DP3O consistently achieves higher accuracy than DPO when used as an implicit preference model. This suggests that DP3O provides a better approximation to the underlying preference structure, which may in turn improve generalization and generation quality. We also observe that DP3O achieves performance comparable to iterative DPO at the final iteration. However, iterative DPO is substantially more expensive (2.67 vs. 1.55 hours) due to the overhead of data generation and additional annotation. Taken together, these results suggest that DP3O offers a favorable trade-off between performance and efficiency.

**Effects of $\alpha$ and preference model type on alignment performance.** We further study how the imitation parameter $\alpha$ and the type of teacher model affect alignment performance. Recall from (8) that a smaller $\alpha$ places more weight on the soft preference probabilities provided by the teacher model, while $\alpha = 1$ reduces DP3O to standard DPO. As shown in the left panel of Figure 3, all main results are obtained using

Table 2: Results on noisy preference data with label-flipping noise ratios of 0.3 and 0.45. **Bold** indicates the best results among all methods.

| Type | Method | $\epsilon = 0.3$ | | | | $\epsilon = 0.45$ | | | |
| | | Win Rate % (↑) | | | $\Delta r$ (↑) | Win Rate % (↑) | | | $\Delta r$ (↑) |
| | | Base | Chosen | Average | | Base | Chosen | Average | |
| Offline | DPO | 65.96 | 32.30 | 49.13 | 0.66 | 52.10 | 23.13 | 37.62 | 0.14 |
| | CDPO | 58.26 | 26.31 | 42.29 | 0.35 | 51.90 | 23.16 | 37.53 | 0.11 |
| | RDPO | 71.30 | 38.71 | 55.01 | 0.89 | 51.67 | 22.42 | 37.05 | 0.11 |
| | d-DPO | 68.37 | 42.14 | 55.26 | 0.91 | 52.39 | 23.27 | 37.83 | 0.16 |
| | DP3O | **71.42** | **47.78** | **59.60** | **1.13** | **54.22** | **25.56** | **39.89** | **0.21** |
| Iterative | Iterative DPO it-1 | 53.42 | 28.27 | 40.85 | 0.30 | 24.42 | 14.25 | 19.34 | $-1.21$ |
| | Iterative DPO it-2 | 57.56 | 30.74 | 44.15 | 0.39 | 18.65 | 6.95 | 12.80 | $-1.51$ |
| | Iterative DPO it-3 | 67.21 | 41.58 | 54.40 | 0.82 | 16.76 | 5.89 | 11.33 | $-1.50$ |

Table 3: AlpacaEval 2 (Li et al., 2023) and MT-Bench (Zheng et al., 2023a) results of two models. **Bold** indicates the best results among all methods.

| Method | Mistral-7B | | | Llama-3-8B | | |
| | AlpacaEval 2 | | MT-Bench | AlpacaEval 2 | | MT-Bench |
| | LC Win Rate (%) | Win Rate (%) | GPT-4 Turbo | LC Win Rate (%) | Win Rate (%) | GPT-4 Turbo |
| SFT | 8.4 | 6.2 | 4.8 | 6.2 | 4.6 | 5.2 |
| SLiC-HF | 10.9 | 8.9 | 5.8 | 12.3 | 13.7 | 6.3 |
| DPO | 15.1 | 12.5 | **5.9** | 18.2 | **15.5** | **6.5** |
| IPO | 11.8 | 9.4 | 5.5 | 14.4 | 14.2 | **6.5** |
| CPO | 9.8 | 8.9 | 5.4 | 10.8 | 8.1 | 6.0 |
| d-DPO | 16.6 | 12.3 | 5.5 | 17.3 | 14.5 | 6.4 |
| DP3O | **17.6** | **13.1** | **5.9** | **19.3** | 15.3 | **6.5** |

an explicit preference model as the default teacher. Under this setting, all choices with $\alpha \neq 1$ outperform $\alpha = 1$, indicating that soft preference information is more effective than hard labels alone. In the right panel of Figure 3, we replace the explicit teacher with an implicit one while fixing $\alpha = 0$ for both cases, and observe worse performance on both average win rate and reward gain $\Delta r$. This suggests that the effectiveness of DP3O depends not only on the use of soft labels, but also on the type of preference model that provides them: explicit preference models offer more effective supervision than implicit ones.

**DP3O consistently outperforms all baselines under label noise.** In addition to the noise-free setting, we also consider label-noise settings. Specifically, we adopt the contamination model $p^\epsilon(y_w \succ y_l \mid x) = (1 - \epsilon) p^\star(y_w \succ y_l \mid x) + \epsilon p^\star(y_l \succ y_w \mid x)$, where $\epsilon$ denotes the noise ratio. We consider two noise levels, $\epsilon = 0.3$ and $\epsilon = 0.45$, with results reported in Table 2. As noise increases, the explicit preference model becomes less accurate, which can degrade iterative DPO because it depends on hard labels from the learned preference model. Offline methods are more robust in comparison. Among them, DP3O performs best: even when the explicit teacher is noisy, its soft probabilities still convey useful uncertainty information, making DP3O more robust than methods based only on hard labels or fixed label smoothing.

In these controlled experiments, we focus on the correctly specified setting, where both the true preference function and the explicit preference model follow the Bradley–Terry form. We consider the misspecified case, where the true preference function is not a BT model, as shown in Appendix A.3.

Table 4: Results on the open LLM leaderboard. **Bold** indicates the best results among all methods.

| Model | Method | MMLU | ARC | HellaSwag | TruthfulQA | Winograd | GSM8K | Average |
|-------|--------|------|-----|-----------|------------|----------|-------|---------|
| | SFT | 60.10 | 58.28 | 80.76 | 40.35 | 76.40 | 28.13 | 57.34 |
| | SLiC-HF | 59.24 | 55.38 | 81.15 | 48.36 | 77.35 | 33.74 | 59.20 |
| | DPO | 58.48 | 61.26 | 83.59 | 53.06 | 76.80 | 21.76 | 59.16 |
| **Mistral 7b** | IPO | 60.23 | 60.84 | 83.30 | 45.44 | 77.58 | 27.14 | 59.09 |
| | CPO | 59.39 | 57.00 | 80.75 | 47.07 | 76.48 | 33.06 | 58.96 |
| | d-DPO | 58.14 | 59.22 | 83.71 | 49.23 | 76.13 | 24.24 | 58.45 |
| | DP3O | 58.13 | 63.16 | 84.76 | 54.23 | 77.59 | 23.53 | **60.23** |
| | SFT | 64.88 | 60.15 | 81.37 | 45.33 | 75.77 | 46.32 | 62.30 |
| | SLiC-HF | 64.36 | 61.43 | 81.88 | 54.95 | 77.27 | 48.82 | 64.79 |
| | DPO | 64.31 | 64.42 | 83.87 | 53.48 | 76.32 | 38.67 | 63.51 |
| **Llama-3-8B** | IPO | 64.40 | 62.88 | 80.46 | 54.20 | 72.22 | 22.67 | 59.47 |
| | CPO | 64.98 | 61.69 | 82.03 | 54.29 | 76.16 | 46.93 | 64.35 |
| | d-DPO | 63.38 | 65.42 | 81.78 | 55.95 | 77.12 | 41.76 | 64.24 |
| | DP3O | 63.18 | 66.30 | 84.01 | 57.42 | 77.43 | 43.29 | **65.27** |

## 5.2 Scaling-up Experiments

**Setup**. We first fine-tuned with two families of models, Mistral-7B (Jiang et al., 2023) and Llama-3-8B (Dubey et al., 2024), on the UltraChat-200k dataset (Ding et al., 2023) to obtain SFT models. Subsequently, we use this SFT model as the starting point for preference optimization and also proxy explicit preference model fitting on the UltraFeedback dataset (Cui et al., 2023).

**Baselines**. In addition to SFT baselines, we compare our method with a range of comprehensive offline alignment approaches: SLiC-HF (Zhao et al., 2023), DPO (Rafailov et al., 2024), IPO (Azar et al., 2024), CPO (Xu et al., 2024a) and d-DPO (Fisch et al., 2024).

**Evaluation.** We evaluate our models using two widely recognized benchmarks: AlpacaEval 2 (Li et al., 2023; Dubois et al., 2024), MT-Bench (Zheng et al., 2023a) and Open LLM Leaderboard (Beeching et al., 2023), which assess a diverse range of conversational abilities across various query types. For AlpacaEval 2, we report both raw win rates and length-controlled (LC) win rates. MT-Bench scores are evaluated with GPT-4-Preview-1106, as it provides more accurate reference answers and judgments compared to GPT-4.

**Results.** Table 3 summarizes the results for Mistral-7B and Llama-3-8B in an offline setup, and here we focus on Mistral-7B. Baseline results are sourced from (Meng et al., 2024). Our proposed method, DP3O, achieves the best or tied-best on 5/6 metrics and remains highly competitive on the remaining one, demonstrating both its effectiveness and robustness for alignment. In particular, for Mistral-7B, DP3O improves the AlpacaEval 2 LC score by 2.5% over the strongest baseline. DP3O also consistently outperforms all other methods across the six downstream tasks listed in Table 4, yielding an average improvement of 2.93% over the base SFT model across two model families. Taken together, these results show that DP3O not only improves alignment performance, but also improves downstream capabilities, making it a strong and practical approach for LLM alignment across diverse models and tasks.

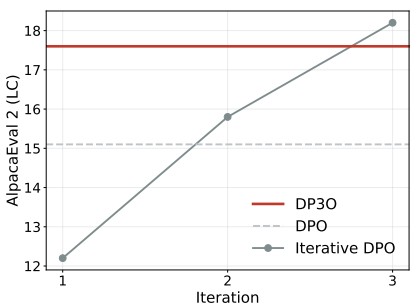

Figure 4: Performance of DP3O, DPO, and iterative DPO on Mistral-7B.

We also compare DP3O with iterative DPO. As shown in Fig. 4, DP3O achieves performance comparable to iterative DPO while requiring only about 50% of the computational time. This result suggests that DP3O captures much of the benefit of iterative preference optimization without incurring its full training cost.

These results highlight DP3O as a performance–efficiency trade-off rather than a method that uniformly dominates iterative alignment. Specifically, DP3O serves as a middle ground between standard offline DPO and iterative DPO: it improves upon DPO through explicit preference modeling while retaining the efficiency and simplicity of offline training. Thus, DP3O narrows the offline–iterative alignment gap, although iterative methods may still achieve stronger task performance in some settings.

## 6 Conclusion

In this work, we investigate the performance gap between offline and iterative alignment approaches by analyzing the role of the preference model in driving their effectiveness. Our controlled experiments demonstrate that the explicit preference model is critical to the performance gap. Motivated by this observation, we propose DP3O, an effective and efficient offline alignment method that leverages knowledge from an explicit preference model while preserving the simplicity of DPO. We evaluate DP3O on a broad set of benchmarks and show that it outperforms state-of-the-art offline methods while narrowing the gap to iterative methods with substantially lower computational cost.

Our findings highlight the potential of offline methods that effectively integrate preference information, offering a promising direction for efficient and scalable alignment of large language models. One limitation of our method is its reliance on training an additional preference model. Future work could address this overhead by exploring lightweight reward models, such as embedding-based approaches (Sun et al., 2025a) or low-rank adapter (LoRA) methods (Zhai et al., 2023) to enable more efficient alignment. Another promising direction is to replace the global interpolation weight $\alpha$ with a sample-adaptive weight $\alpha_i$, relying less on the explicit preference model on instances where it is unreliable.

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

# A   Additional Results

## A.1   Details about Controlled Experiments

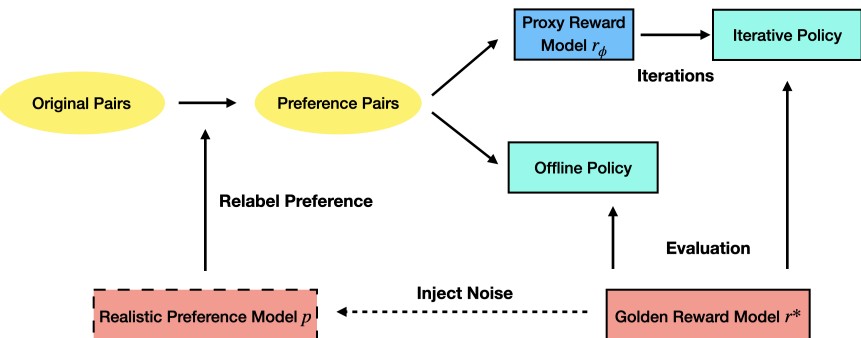

Figure 5: The overall framework for controlled experiments comparing iterative and offline approaches.

**Algorithms.** We compare DP3O with a diverse set of alignment baselines, including several representative *offline* methods and one *iterative* method. Specifically, the offline baselines include DPO (Rafailov et al., 2024), its label-smoothing variants CDPO (Mitchell, 2023) and RDPO (Chowdhury et al., 2024), and the reward-distillation method d-DPO (Fisch et al., 2024). We also include iterative DPO as a strong iterative alignment baseline.

For a fair comparison, all policy models are instantiated with Pythia-2.8B. For CDPO and RDPO, we set the label-smoothing parameter to 0.3, which is intended to reflect the stochasticity of the preference labels in the dataset. For d-DPO, we first train an explicit preference model on the pairwise preference data, and then use its output logits as training targets for the implicit reward model, optimized with a mean squared error objective. For iterative DPO, we run 3 iterations in total. At each iteration, we generate 8 candidate responses per prompt and use one-third of the prompts, so that the total prompt budget is matched across iterations.

**Data.** We conduct experiments on the Anthropic Helpful and Harmless (HH) dialogue dataset (Bai et al., 2022). This dataset consists of prompts paired with two model-generated responses, together with a human preference label indicating which response is preferred. Following common practice, we retain only the single-turn dialogue instances to reduce confounding factors introduced by longer multi-turn interactions. The resulting dataset is split into training, validation, and test sets containing 4.7k, 1k, and 2.5k prompts, respectively.

**Models.** We use Pythia-2.8B (Biderman et al., 2023) as the policy model in all experiments. We first perform supervised fine-tuning on the preferred completions in the training set to obtain the reference policy $\pi_{\mathrm{ref}}$, which serves as the initialization and reference model for subsequent preference optimization.

For the gold reward model $r^\star$, we train a Mistral-7B-Instruct-v0.2 model (Jiang et al., 2023) on the corresponding preference pairs. This reward model is used only for evaluation and serves as an external judge for comparing generations from different policies.

For the explicit preference model, we adopt the same backbone architecture as the policy model, i.e., Pythia-2.8B. To convert the language model into an explicit preference model that outputs a scalar logit, we replace the final language modeling head with a linear regression head that predicts a single scalar reward score for each response, following prior work on reward modeling and preference learning (Dong et al., 2024). The resulting model is trained on pairwise preference data and is later used to produce the soft preference probabilities required by DP3O.

**Hyperparameters.** For policy optimization, we use the same hyperparameter setting across all methods unless otherwise specified. In particular, we set $\beta = 0.05$, the learning rate to $1 \times 10^{-6}$, and the batch size to 64. All policy optimization experiments are run for a single epoch.

For training the explicit preference model, we use a batch size of 128 and a learning rate of $1 \times 10^{-5}$. For the gold reward model, we use a batch size of 256 and a learning rate of $5 \times 10^{-6}$. All reward and preference models are also trained for a single epoch.

For CDPO and RDPO, the label-smoothing factor is set to 0.3. This value is chosen because it approximately matches the underlying label-flip rate induced by the ground-truth preference distribution in our synthetic setup, and thus provides these methods with a well-calibrated smoothing level.

**Evaluation.** We evaluate all models using two metrics derived from the gold reward model $r^\star$: *win rate* and *reward improvement* ($\Delta$). The win rate measures the fraction of test prompts for which the response generated by the current policy $\pi_\theta$ receives a higher gold reward than the response generated by the reference policy $\pi_{\mathrm{ref}}$. The reward improvement metric measures the average difference in gold reward between these two responses.

Formally, the metrics are defined as

$$\text{Win Rate} = \frac{1}{|\mathcal{X}_{\mathrm{test}}|} \sum_{x \in \mathcal{X}_{\mathrm{test}}} \mathbb{1}\big[r^\star(x, y_{\pi_\theta}) > r^\star(x, y_{\pi_{\mathrm{ref}}})\big],$$

$$\text{Reward } \Delta = \frac{1}{|\mathcal{X}_{\mathrm{test}}|} \sum_{x \in \mathcal{X}_{\mathrm{test}}} \big[r^\star(x, y_{\pi_\theta}) - r^\star(x, y_{\pi_{\mathrm{ref}}})\big],$$

where $\mathcal{X}_{\mathrm{test}}$ denotes the held-out test prompt set, and $y_{\pi_\theta}$ and $y_{\pi_{\mathrm{ref}}}$ are the responses sampled from $\pi_\theta$ and $\pi_{\mathrm{ref}}$, respectively. During evaluation, we generate responses using stochastic decoding with temperature 0.7.

**Compute.** All training and inference experiments are conducted on four NVIDIA A6000 GPUs with 40GB memory each.

## A.2 Details about Scaling Experiments

**Baselines.** We compare DP3O against several strong preference optimization baselines, including SLiC-HF (Zhao et al., 2023), DPO (Rafailov et al., 2024), IPO (Azar et al., 2024), and CPO (Xu et al., 2024a). For a fair comparison, we only include baselines that do not rely on length normalization tricks. Extending DP3O with such techniques is left for future work.

**Hyperparameters.** For Mistral models, we set $\beta = 0.01$ and use a learning rate of $5 \times 10^{-7}$. For Llama-3-8B models, we set $\beta = 0.01$ and use a learning rate of $6 \times 10^{-7}$. The batch size for policy optimization is 32, and all models are trained for one epoch. For fitting the explicit preference model, we use a learning rate of $5 \times 10^{-6}$ and a batch size of 256.

**Evaluation.** We evaluate our models on AlpacaEval 2, MT-Bench, and the Open LLM Leaderboard.

For AlpacaEval 2, we follow the standard evaluation protocol and report the length-controlled win rate. During evaluation, we use sampling-based decoding, with temperature 0.7 for Mistral-7B and 0.9 for Llama-3-8B.

For MT-Bench, we follow the official setting and report the average score over all questions. We use the benchmark's default category-specific decoding temperatures.

For the Open LLM Leaderboard, we report the average score across six standard tasks: ARC, HellaSwag, MMLU, TruthfulQA, Winogrande, and GSM8K, which together evaluate science question answering, commonsense reasoning, multitask knowledge, truthfulness, and mathematical reasoning, following prior work.

**Compute.** All training and inference experiments are conducted on four NVIDIA A6000 GPUs with 40GB memory each.

## A.3 Misspecified Explicit Preference model

DP3O relies on an explicit preference model to provide soft preference probabilities for distillation. Ideally, this model should match the underlying true preference function. In practice, however, this assumption

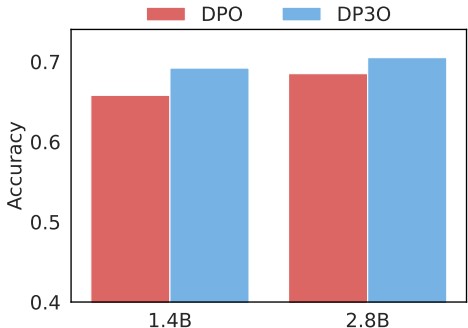

Figure 6: Accuracy of implicit models.

Table 5: Comparison of the explicit preference model and LLM-as-judge baselines.

| Method | ACC (%) (↑) | ECE (↓) | Win rate (↑) |
|---|---|---|---|
| Explicit | **82.4** | **0.24** | **77.1** |
| LLM-judge (prompt) | 52.2 | 0.46 | 70.4 |
| LLM-judge (trained) | 68.3 | 0.32 | 74.8 |

may not hold. For example, the true preference function may be induced by an LLM-as-judge, while the explicit preference model used by DP3O is trained separately on offline preference data. A natural question is therefore whether the explicit preference model remains useful even when it is misspecified relative to the true preference function.

To study this setting, we conduct an experiment in which the ground-truth preference function is defined by an LLM-as-judge, specifically Mistral-7B-SFT-beta. We then compare our explicit preference model with two LLM-as-judge alternatives. The first is a *prompt-based* judge, where an instruction-tuned model is prompted to choose between two responses, and the preference probability is estimated from the token probabilities of "A" and "B". The second is a *trained* judge, where the same backbone is further supervised fine-tuned for pairwise judgment. We use Gemma-2B-it as the backbone for judge baselines, since it is already instruction-tuned and better suited for following judgment prompts. We use $\alpha = 0$ for DP3O.

We evaluate these models using three metrics. *Accuracy (ACC)* measures how often the model correctly predicts the preferred response. *Expected calibration error (ECE)* measures the discrepancy between predicted preference probabilities and empirical correctness, with lower values indicating better calibration. Finally, *win rate* measures downstream policy performance when DP3O uses each model as the preference model, reporting how often the resulting policy is preferred over the reference policy under the evaluation setup.

The results are shown in Table 5. Even though the explicit preference model is misspecified in this setting, it still achieves the best overall trade-off between accuracy and calibration, and DP3O built on top of it attains the highest win rate. By contrast, the prompt-based LLM-as-judge performs poorly in both accuracy and calibration, while supervised fine-tuning improves the LLM-as-judge baseline but still leaves it substantially less calibrated than the explicit model.

These results suggest that an explicit preference model can remain highly effective for DP3O even when it does not exactly match the true preference function. What matters is not only whether the model is misspecified, but whether it can still provide accurate and well-calibrated soft preference probabilities. In our setting, the explicit preference model appears to be a more reliable source of such signals than LLM-as-judge alternatives, leading to stronger downstream policy optimization.

### A.4 Why the Implicit Class $\Pi$ is Richer than the Explicit Class $\mathcal{P}$?

Theorem 4.1 relies on the size of the hypothesis class. Since analytically comparing two deep hypothesis classes that share a backbone is intractable, we provide two complementary lines of evidence that the implicit family parameterizes a strictly larger set of preference functions: (i) a *structural* argument showing that the implicit read-out carries far more parameters and a far higher-dimensional output space than the explicit read-out, and (ii) an *empirical* argument, in the spirit of Zhang et al. (2017), showing that this larger parameterization translates into substantially greater expressive power, i.e., the ability to memorize arbitrary random preference labels that the explicit family cannot fit.

**Different parameters in the output layer.** With a shared backbone of hidden size $d$ and vocabulary size $V$, the two families differ only at the output layer, and the implicit family has strictly more parameters there:

- **Explicit model:** a linear reward head $\mathbb{R}^d \to \mathbb{R}$ with $d$ parameters, collapsing every response to a single scalar $r_\phi(x, y) \in \mathbb{R}$.

- **Implicit model:** the LM head $\mathbb{R}^d \to \mathbb{R}^V$ with $d \times V$ parameters, and the reward is realized as a sum of per-token log-ratios

$$r_\theta(x, y) \;=\; \beta \sum_{t=1}^{T} \log \frac{\pi_\theta(y_t \mid x, y_{<t})}{\pi_{\text{ref}}(y_t \mid x, y_{<t})}. \tag{9}$$

Instantiating these counts for Pythia-1.4B ($d = 2048$, $V = 50304$, with $T$ typically in the hundreds), the explicit reward head carries about **2K** parameters and produces a 1-dim output per response, while the implicit LM-head read-out carries about **103M** parameters and produces an effective $V \times T$-dim output per response on the order of tens of millions. The two output heads thus differ by roughly five orders of magnitude in parameter count and seven orders of magnitude in effective output dimensionality, even though the upstream backbone is identical. The extra $V \times d$ parameters at the read-out are not just a parameter-count difference: they let the implicit model assign an independent log-ratio to every (token, position) pair, so the response-level reward is a sum of $T$ unbounded log-ratios, versus a single bounded scalar in the explicit case. The implicit parameterization therefore indexes a strictly larger family of preference functions.

**Empirical capacity measurement via the randomization test.** The structural argument bounds the family from above; we next show that the implicit family can realize this extra flexibility on data. Following the randomization-test protocol of Zhang et al. (2017), a more flexible parameterization should be able to fit signal-free random preference labels more easily, even when the backbone, optimizer, and regularization are held identical. This isolates the contribution of the output parameterization from the shared backbone.

We vary *only* the output parameterization, holding everything else fixed. We start from the Anthropic HH-RLHF preference dataset. To remove learnable preference signals from training, we randomly assign the preference label for each training pair independently of the original annotation; that is, for each pair, either response is selected as the winner with probability $1/2$. The validation set is left intact with the original labels, so that the evaluation signal remains a clean probe of whether the model has learned transferable preferences or simply fit the randomized training labels. Both families are initialized from the same Pythia-1.4B SFT checkpoint. The only varied factor is the output parameterization: the explicit model optimizes $\mathcal{L}_{\text{Explicit}} = -\log \sigma\big(\beta\left(r_\phi(x, y_w) - r_\phi(x, y_l)\right)\big)$, where $r_\phi$ is a linear scalar head on the final hidden state, swept over $\beta \in \{0.05, 0.1, 0.2, 1.0\}$; the implicit DPO model optimizes $\mathcal{L}_{\text{DPO}} = -\log \sigma\big(\beta(\log \frac{\pi_\theta(y_w|x)}{\pi_{\text{ref}}(y_w|x)} - \log \frac{\pi_\theta(y_l|x)}{\pi_{\text{ref}}(y_l|x)})\big)$ with $\beta \in \{0.05, 0.1, 0.2\}$. We almost match $\beta$ across the two families so that any difference cannot be attributed to logit-scale rescaling. All runs use the same optimizer, learning rate $1 \times 10^{-5}$ and cosine schedule with warmup ratio 0.1, trained for 3 epochs at batch size 64.

Figure 7 reports train and eval loss across all $\beta$ values. On the randomized-label training set, the explicit model's train loss stays near the random-guessing level for every $\beta$, indicating that it does not fit the randomly assigned labels. In contrast, Implicit DPO drives the train loss close to zero, achieving near-perfect memorization of the randomized training preferences. On the clean validation set, the explicit model's eval

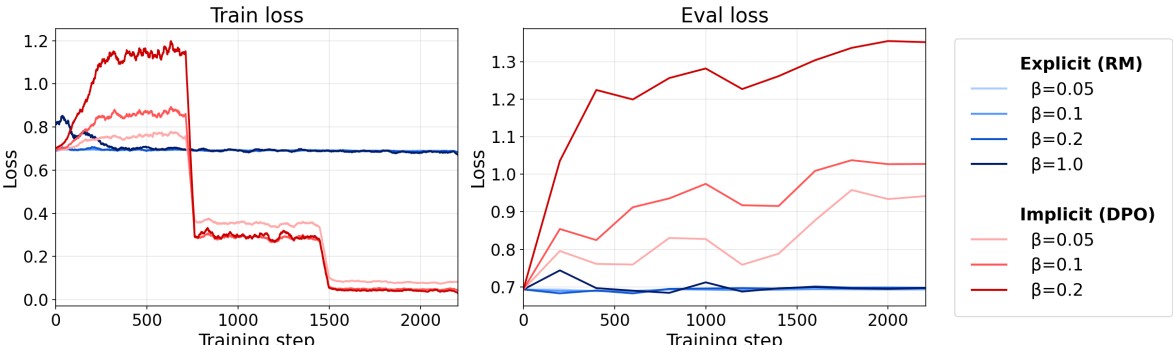

Figure 7: Randomization test on Pythia-1.4B with randomly assigned training preference labels. **Left:** Train loss on the randomized-label training set. Explicit (blue, all $\beta$) plateaus near the random-guessing loss and does not fit the randomly assigned labels. Implicit DPO (red, all $\beta$) drives the train loss close to zero. **Right:** Eval loss on the clean validation set with original labels. Explicit stays near the random-guessing loss, while the eval loss of Implicit DPO increases, showing that it fits the randomized training labels rather than learning transferable preference structure. The gap is consistent across $\beta$ values, isolating the contribution of the output parameterization from the shared backbone.

loss also stays near the random-guessing level, consistent with learning no transferable preference structure. By contrast, Implicit DPO's eval loss increases substantially, indicating that it has fit the randomized training labels and moved away from the original preference signal. This behavior is a direct fingerprint of memorizing randomized labels rather than learning genuine preference structure. The separation is consistent across $\beta$ values with identical backbone, data, and regularization.

Taken together, the structural argument and the randomization test suggest that, in our matched training setup, the implicit parameterization provides a more flexible way to fit preference labels and is more prone to memorization than the explicit Bradley–Terry parameterization.

### A.5 Ablation on Preference Signals in Iterative methods

To further understand the role of explicit preference modeling and soft preference distillation in iterative alignment, we compare three iterative variants: iterative DPO with a separately trained explicit preference model, iterative DPO with an implicit preference model induced by the policy, and iterative DP3O, which uses the soft preference probabilities produced by the explicit preference model. All methods follow the same iterative training pipeline and differ only in how the preference signal for newly generated responses is constructed. As shown in Figure 8, iterative DPO with the explicit preference model consistently outperforms the variant using the implicit preference model across all iterations. At the final iteration, iterative DPO with the explicit preference model achieves a win rate of 72.69%, compared with 66.76% for the implicit variant. This suggests that the explicit preference model provides a stronger signal for selecting and labeling generated responses, and that the benefit of iterative DPO is not solely due to iterative data generation.

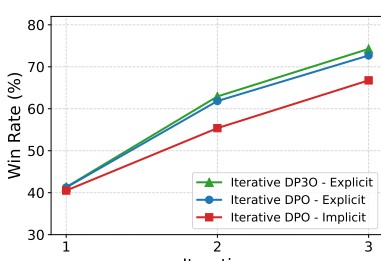

Figure 8: Ablation on preference signals in iterative alignment. Iterative DPO with an explicit preference model outperforms its implicit counterpart, while iterative DP3O further improves performance by using explicit soft probabilities.

Moreover, iterative DP3O further improves the final win rate to 74.25% by using the soft preference probabilities from the same explicit preference model rather than converting them into hard winner–loser labels. These results indicate that explicit preference modeling is an important contributor to the offline–iterative gap, and that soft preference distillation can further leverage the information captured by the explicit preference model.

# B Proofs

## B.1 Estimation Error of Preference Model

**Theorem B.1.** *Let $p^\star(y_1 \succ y_2 \mid x)$ denote the true pairwise preference probability for $x \in \mathcal{X}$ and $y_1, y_2 \in \mathcal{Y}$. Let $\rho \in \Delta(\mathcal{X})$ be a prompt distribution and $\pi(\cdot \mid x) \in \Delta(\mathcal{Y})$ a response distribution. For each $i \in [n]$, sample $x^{(i)} \sim \rho$ and $y_1^{(i)}, y_2^{(i)} \stackrel{\text{i.i.d.}}{\sim} \pi(\cdot \mid x^{(i)})$, then draw a preference according to $p^\star(y_1^{(i)} \succ y_2^{(i)} \mid x^{(i)})$. Let $(y_w^{(i)}, y_l^{(i)})$ be the induced winner-loser ordering, and define $\mathcal{D} = \{(x^{(i)}, y_w^{(i)}, y_l^{(i)})\}_{i=1}^n$.*

*Let $\mathcal{F}$ be a finite class of functions $p : \mathcal{X} \times \mathcal{Y} \times \mathcal{Y} \to [0, 1]$ such that $p^\star \in \mathcal{F}$, and assume that $0 < p(y_1 \succ y_2 \mid x) < 1$ for all $p \in \mathcal{F}$ and all $(x, y_1, y_2) \in \mathcal{X} \times \mathcal{Y} \times \mathcal{Y}$. Let*

$$\widehat{p} := \arg\max_{p \in \mathcal{F}} \sum_{(x, y_w, y_l) \in \mathcal{D}} \log p(y_w \succ y_l \mid x).$$

*Then for any $\delta \in (0, 1)$, with probability at least $1 - \delta$,*

$$\mathbb{E}_{x \sim \rho, \, y_1, y_2 \sim \pi(\cdot \mid x)} \left[ D_{\mathrm{H,Ber}}^2 \big(\hat{p}(y_1 \succ y_2 \mid x), \, p^\star(y_1 \succ y_2 \mid x)\big) \right] \le \frac{2 \log(|\mathcal{F}|/\delta)}{n},$$

*where $D_{\mathrm{H,Ber}}^2(a, b) := D_{\mathrm{H}}^2(\mathrm{Ber}(a), \mathrm{Ber}(b))$ and $D_{\mathrm{H}}^2(P, Q) := \int \big(\sqrt{p(z)} - \sqrt{q(z)}\big)^2 dz$.*

We present a self-contained proof for the finite-class preference modeling setting. The proof is inspired by the likelihood-ratio and Hellinger-affinity arguments underlying Zhang (2006), but it does not directly invoke the full information-complexity theorem of Zhang (2006). While Zhang's general framework includes an explicit KL-entropy complexity penalty, in our finite-class setting with a uniform prior this penalty is constant across all deterministic estimators. Indeed, for any estimator selecting $p \in \mathcal{F}$, we have $D_{\mathrm{KL}}(\delta_p \| \pi) = \log |\mathcal{F}|$, which is independent of $p$. Hence, adding this penalty does not change the optimizer, and the corresponding complexity-regularized estimator coincides with the standard MLE considered here.

For each $p \in \mathcal{P}$, define the conditional Bernoulli model

$$q_p(b \mid x, y_1, y_2) := p(y_1 \succ y_2 \mid x)^b \big(1 - p(y_1 \succ y_2 \mid x)\big)^{1-b}, \qquad b \in \{0, 1\},$$

and similarly

$$q^\star(b \mid x, y_1, y_2) := \big(p^\star(y_1 \succ y_2 \mid x)\big)^b \big(1 - p^\star(y_1 \succ y_2 \mid x)\big)^{1-b}.$$

**Lemma B.2** (Hellinger transform identity)**.** *For any $p \in \mathcal{P}$,*

$$\mathbb{E}\left[ \sqrt{\prod_{i=1}^n \frac{q_p\big(b^{(i)} \mid x^{(i)}, y_1^{(i)}, y_2^{(i)}\big)}{q^\star\big(b^{(i)} \mid x^{(i)}, y_1^{(i)}, y_2^{(i)}\big)}} \right] = \big(1 - \Delta(p)\big)^n,$$

*where*

$$\Delta(p) := \mathbb{E}_{x \sim \rho, \, y_1, y_2 \sim \pi(\cdot \mid x)} \left[ D_{\mathrm{H,Ber}}^2 \big(p(y_1 \succ y_2 \mid x), p^\star(y_1 \succ y_2 \mid x)\big) \right].$$

*Proof of lemma.* By independence,

$$\mathbb{E}\left[ \sqrt{\prod_{i=1}^n \frac{q_p\big(b^{(i)} \mid x^{(i)}, y_1^{(i)}, y_2^{(i)}\big)}{q^\star\big(b^{(i)} \mid x^{(i)}, y_1^{(i)}, y_2^{(i)}\big)}} \right]$$
$$= \prod_{i=1}^n \mathbb{E}\left[ \sqrt{\frac{q_p\big(b^{(i)} \mid x^{(i)}, y_1^{(i)}, y_2^{(i)}\big)}{q^\star\big(b^{(i)} \mid x^{(i)}, y_1^{(i)}, y_2^{(i)}\big)}} \right].$$

Since the samples are identically distributed, it suffices to evaluate one factor:

$$\mathbb{E}\left[\sqrt{\frac{q_p(b \mid x, y_1, y_2)}{q^\star(b \mid x, y_1, y_2)}}\right]$$

$$= \mathbb{E}_{x \sim \rho, \, y_1, y_2 \sim \pi(\cdot \mid x)}\left[\sum_{b \in \{0,1\}} q^\star(b \mid x, y_1, y_2)\sqrt{\frac{q_p(b \mid x, y_1, y_2)}{q^\star(b \mid x, y_1, y_2)}}\right]$$

$$= \mathbb{E}_{x \sim \rho, \, y_1, y_2 \sim \pi(\cdot \mid x)}\left[\sum_{b \in \{0,1\}} \sqrt{q_p(b \mid x, y_1, y_2)\, q^\star(b \mid x, y_1, y_2)}\right].$$

For Bernoulli distributions, the inner term equals

$$1 - D_{\mathrm{H,Ber}}^2\big(p(y_1 \succ y_2 \mid x), p^\star(y_1 \succ y_2 \mid x)\big).$$

Taking expectation yields $1 - \Delta(p)$, and taking the product over $i = 1, \ldots, n$ proves the claim. $\qquad\square$

*Proof.* For each $i \in [n]$, let

$$b^{(i)} := \mathbf{1}\{y_1^{(i)} \succ y_2^{(i)}\}.$$

Then the tuples

$$\big(x^{(i)}, y_1^{(i)}, y_2^{(i)}, b^{(i)}\big), \qquad i = 1, \ldots, n,$$

are i.i.d., with

$$x^{(i)} \sim \rho, \qquad y_1^{(i)}, y_2^{(i)} \overset{\text{i.i.d.}}{\sim} \pi(\cdot \mid x^{(i)}), \qquad b^{(i)} \sim q^\star(\cdot \mid x^{(i)}, y_1^{(i)}, y_2^{(i)}).$$

Moreover, by definition of the winner-loser ordering,

$$\sum_{(x, y_w, y_l) \in \mathcal{D}} \log p(y_w \succ y_l \mid x) = \sum_{i=1}^{n} \log q_p\big(b^{(i)} \mid x^{(i)}, y_1^{(i)}, y_2^{(i)}\big).$$

Hence

$$\widehat{p} = \arg\max_{p \in \mathcal{P}} \sum_{i=1}^{n} \log q_p\big(b^{(i)} \mid x^{(i)}, y_1^{(i)}, y_2^{(i)}\big).$$

Now fix any $p \in \mathcal{P}$. If

$$\sum_{i=1}^{n} \log q_p\big(b^{(i)} \mid x^{(i)}, y_1^{(i)}, y_2^{(i)}\big) \geq \sum_{i=1}^{n} \log q^\star\big(b^{(i)} \mid x^{(i)}, y_1^{(i)}, y_2^{(i)}\big),$$

then

$$\prod_{i=1}^{n} \frac{q_p\big(b^{(i)} \mid x^{(i)}, y_1^{(i)}, y_2^{(i)}\big)}{q^\star\big(b^{(i)} \mid x^{(i)}, y_1^{(i)}, y_2^{(i)}\big)} \geq 1,$$

and hence

$$\sqrt{\prod_{i=1}^{n} \frac{q_p\big(b^{(i)} \mid x^{(i)}, y_1^{(i)}, y_2^{(i)}\big)}{q^\star\big(b^{(i)} \mid x^{(i)}, y_1^{(i)}, y_2^{(i)}\big)}} \geq 1.$$

Therefore, by Markov's inequality and the Hellinger transform identity,

$$\mathbb{P}\left(\sum_{i=1}^{n} \log q_p\big(b^{(i)} \mid x^{(i)}, y_1^{(i)}, y_2^{(i)}\big) \geq \sum_{i=1}^{n} \log q^\star\big(b^{(i)} \mid x^{(i)}, y_1^{(i)}, y_2^{(i)}\big)\right)$$

$$\leq \mathbb{E}\left[\sqrt{\prod_{i=1}^{n} \frac{q_p\big(b^{(i)} \mid x^{(i)}, y_1^{(i)}, y_2^{(i)}\big)}{q^\star\big(b^{(i)} \mid x^{(i)}, y_1^{(i)}, y_2^{(i)}\big)}}\right]$$

$$= (1 - \Delta(p))^n \leq e^{-n\Delta(p)},$$

where the last step uses $1 - u \le e^{-u}$.

For $\varepsilon > 0$, define

$$\mathcal{P}_\varepsilon := \{p \in \mathcal{P} : \Delta(p) > \varepsilon\}.$$

Since $p^\star \in \mathcal{P}$ and $\widehat{p}$ maximizes the empirical log-likelihood over $\mathcal{P}$, the event $\{\Delta(\widehat{p}) > \varepsilon\}$ implies that there exists some $p \in \mathcal{P}_\varepsilon$ such that

$$\sum_{i=1}^{n} \log q_p\big(b^{(i)} \mid x^{(i)}, y_1^{(i)}, y_2^{(i)}\big) \ge \sum_{i=1}^{n} \log q^\star\big(b^{(i)} \mid x^{(i)}, y_1^{(i)}, y_2^{(i)}\big).$$

Therefore, by the union bound,

$$\mathbb{P}\big(\Delta(\widehat{p}) > \varepsilon\big) \le \sum_{p \in \mathcal{P}_\varepsilon} \mathbb{P}\left(\sum_{i=1}^{n} \log q_p\big(b^{(i)} \mid x^{(i)}, y_1^{(i)}, y_2^{(i)}\big) \ge \sum_{i=1}^{n} \log q^\star\big(b^{(i)} \mid x^{(i)}, y_1^{(i)}, y_2^{(i)}\big)\right) \le |\mathcal{P}_\varepsilon| e^{-n\varepsilon} \le |\mathcal{P}| e^{-n\varepsilon}.$$

Setting $\varepsilon = \frac{2\log(|\mathcal{P}|/\delta)}{n}$, it gives $\mathbb{P}\big(\Delta(\widehat{p}) > \varepsilon\big) \le |\mathcal{P}| e^{-n\varepsilon} = |\mathcal{P}| e^{-2\log(|\mathcal{P}|/\delta)} = |\mathcal{P}| \left(\frac{\delta}{|\mathcal{P}|}\right)^2 \le \delta.$.

Hence, with probability at least $1 - \delta$,

$$\Delta(\widehat{p}) \le \frac{2\log(|\mathcal{P}|/\delta)}{n}.$$

Substituting the definition of $\Delta(\widehat{p})$ completes the proof. $\qquad\square$

## B.2 Understanding the Benefits of Preference Distillation via Risk Decomposition

We build upon the theoretical result based on (Menon et al., 2021), extending it from the original classification to RLHF setting.

**Lemma B.3.** *For any policy $\pi_t \in \Pi$ and $\mathcal{D}_t \sim \mathbb{P}_{\mathcal{D}_t}$ :*

$$\mathbb{V}_{\mathcal{D}_t} \hat{R}_{DP3O}(\pi_t; \mathcal{D}_t, p^*) \le \mathbb{V}_{\mathcal{D}_t} \hat{R}_{DPO}(\pi_t; \mathcal{D}_t).$$

*Proof.* We first denote the preference label $l$ and further define $p(l = 1 \mid x, y_w, y_l) = p(y_w \succ y_l \mid x)$ and $p(l = 0 \mid x, y_w, y_l) = p(y_w \prec y_l \mid x)$

$$
\begin{aligned}
\mathbb{V}_{\mathcal{D}_t} \hat{R}_{\text{DP3O}}\left(\pi_t; \mathcal{D}_t, p^*\right) &= \mathbb{V}_{\mathcal{D}_t} \left[ -\frac{1}{n_t} \sum_{i=1}^{n_t} \left[ \sum_l p\left(l^{(i)} \mid x^{(i)}, y_w^{(i)}, y_l^{(i)}\right) \log p_\pi\left(l^{(i)} \mid x^{(i)}, y_w^{(i)}, y_l^{(i)}\right) \right] \right] \\
&= \frac{1}{n_t} \mathbb{V} \left[ \sum_l p\left(l^{(i)} \mid x^{(i)}, y_w^{(i)}, y_l^{(i)}\right) \log p_\pi\left(l^{(i)} \mid x^{(i)}, y_w^{(i)}, y_l^{(i)}\right) \right] \\
&= \frac{1}{n_t} \mathbb{V} \left[ \mathbb{E}_{l \mid x, y_w, y_l} \left[\log p_\pi\left(l \mid x, y_w, y_l\right)\right] \right] \\
&= \frac{1}{n_t} \left( \mathbb{E}\left[ \mathbb{E}_{l \mid x, y_w, y_l} \left[\log p_\pi\left(l \mid x, y_w, y_l\right)\right] \right]^2 - \left[ \mathbb{E}_{l \mid x, y_w, y_l} \left[\log p_\pi\left(l \mid x, y_w, y_l\right)\right] \right]^2 \right) \\
&\le \frac{1}{n_t} \left( \mathbb{E}\left[ \mathbb{E}_{l \mid x, y_w, y_l} \left[\log p_\pi\left(l \mid x, y_w, y_l\right)\right]^2 \right] - \left[ \mathbb{E}_{l \mid x, y_w, y_l} \left[\log p_\pi\left(l \mid x, y_w, y_l\right)\right] \right]^2 \right) \\
&= \frac{1}{n_t} \mathbb{V} \left[\log p_\pi\left(l \mid x, y_w, y_l\right)\right] \\
&= \mathbb{V}_{\mathcal{D}_t} \hat{R}_{\text{DPO}}\left(\pi_t; \mathcal{D}_t\right),
\end{aligned}
$$

where the inequality follows from Jensen's inequality. $\qquad\square$

**Theorem B.4.** *For any policy $\pi_t \in \Pi$ and $\hat{p} \in \mathcal{P}$, Then, for any $\delta \in (0,1)$, with probability at least $1 - \delta$ over $\mathcal{D}_t \sim \mathbb{P}_{\mathcal{D}_t}$ :*

$$R(\pi_t) \le \hat{R}_{DP3O}(\pi_t; \mathcal{D}_t, \hat{p}) + \mathcal{O}\left(\sqrt{\hat{\mathbb{V}}_{\mathcal{D}_t}(\pi_t, \hat{p}) \cdot \frac{\log \frac{|\Pi|}{\delta}}{n_t}} + \frac{\log |\Pi|}{n_t}\right) + \mathcal{O}\left(\mathbb{E}_{\mathcal{D}_t} \|\hat{p} - p^*\|_2\right).$$

*Proof.* Let $R(\pi_t; \hat{p}) = \mathbb{E}\left[\hat{R}_{\mathrm{DP3O}}(\pi_t; \mathcal{D}_t, \hat{p})\right]$. Then we have

$$
\begin{aligned}
|R(\pi_t; \hat{p}) - R(\pi_t)| &= \left|\mathbb{E}\left[\hat{R}_{\mathrm{DP3O}}(\pi_t; \mathcal{D}_t, \hat{p})\right] - \mathbb{E}\left[\hat{R}_{\mathrm{DP3O}}(\pi_t; \mathcal{D}_t, p^*)\right]\right| \\
&= \left|\mathbb{E}\left[\hat{p}(x, y_l, y_w)^\top \log p_{\pi_t}(x, y_l, y_w) - p^*(x, y_l, y_w)^\top \log p_{\pi_t}(x, y_l, y_w)\right]\right| \\
&\le \mathbb{E}\left[\left|\hat{p}(x, y_l, y_w)^\top \log p_{\pi_t}(x, y_l, y_w) - p^*(x, y_l, y_w)^\top \log p_{\pi_t}(x, y_l, y_w)\right|\right] \\
&= \mathbb{E}\left[\left|\left(\hat{p}(x, y_l, y_w)^\top - p^*(x, y_l, y_w)^\top\right) \log p_{\pi_t}(x, y_l, y_w)\right|\right] \\
&\le \mathbb{E}\left[\|\hat{p} - p^*\|_2 \cdot \|\log p_{\pi_t}(x, y_l, y_w)\|_2\right] \\
&\le \mathbb{E}\left[\|\hat{p} - p^*\|_2 \cdot \sqrt{\log^2(\tau) + \log^2(1 - \tau)}\right] \\
&= C \cdot \mathbb{E}\left[\|\hat{p} - p^*\|_2\right].
\end{aligned}
$$

where $C = \sqrt{\log^2(\tau) + \log^2(1 - \tau)}$. The second inequality follows from the Cauchy-Schwarz inequality, and the final inequality is a consequence of the boundedness of $p_\pi$. We can directly get $R(\pi_t) \le R(\pi_t; \hat{p}) + C \cdot \mathbb{E}\left[\|\hat{p} - p^*\|_2\right]$ following the last equality.

Using Lemma B.5, we can obtain that with probability $1 - \delta$:

$$R(\pi_t; \hat{p}) \le \hat{R}_{\mathrm{DP3O}}(\pi_t; \mathcal{D}_t, \hat{p}) + \mathcal{O}\left(\sqrt{\hat{\mathbb{V}}_{\mathcal{D}_t}(\pi_t, \hat{p}) \cdot \frac{\log \frac{|\Pi|}{\delta}}{n_t}} + \frac{\log \frac{|\Pi|}{\delta}}{n_t}\right) \tag{10}$$

where $\hat{\mathbb{V}}_{\mathcal{D}_t}(\pi_t, \hat{p})$ is the empirical variance of the DP3O loss.

Then we have

$$R(\pi_t) \le R(\pi_t; \hat{p}) + C \cdot \mathbb{E}\left[\|\hat{p} - p^*\|_2\right].$$

$$\le \hat{R}(\pi_t; \mathcal{D}_t, \hat{p}) + \mathcal{O}\left(\sqrt{\hat{\mathbb{V}}_{\mathcal{D}_t}(\pi_t, \hat{p}) \cdot \frac{\log \frac{|\Pi|}{\delta}}{n_t}} + \frac{\log \frac{|\Pi|}{\delta}}{n_t}\right) + \mathcal{O}\left(\mathbb{E}\left[\|\hat{p} - p^*\|_2\right]\right).$$

The last inequality directly follows (10). $\qquad\qquad\square$

**Lemma B.5.**

$$R(\pi_t; \hat{p}) \le \hat{R}(\pi_t; \mathcal{D}_t, \hat{p}) + \mathcal{O}\left(\sqrt{\hat{\mathbb{V}}_{\mathcal{D}_t}(\pi_t, \hat{p}) \cdot \frac{\log \frac{|\Pi|}{\delta}}{n_t}} + \frac{\log \frac{|\Pi|}{\delta}}{n_t}\right) \tag{11}$$

The Lemma mainly adopted Corollary 5 from (Maurer & Pontil, 2009), so we omitted the proof here.

Figure 9: Two AlpacaEval 2 generations from the Mistral model after training with DPO and DP3O.

**(a) DPO**

| | |
|---|---|
| Input → | Q: How can you determine if a person is genuinely interested in a conversation or simply being polite? |

| | |
|---|---|
| Mistral DPO → | As an AI, I don't have access to contextual information or nonverbal cues, but here are some general signs that may indicate whether someone is genuinely interested in a conversation or being polite:
1. Active listening: A genuinely interested person will actively listen to what you're saying, ask questions, and provide feedback or comments. They may also maintain eye contact, nod, or use verbal cues like "uh-huh" or "I see" to show they're engaged.
2. Follow-up questions: A genuine interest in the conversation often leads to follow-up questions. If someone is asking questions that show they're interested in learning more about the topic, it's a good sign they're engaged.
3. Body language: Nonverbal cues like leaning in, smiling, and mirroring your body language can indicate genuine interest. However, note that some people may have different body language styles, so this may not always be a reliable indicator.
4. Topic continuation: If the conversation flows naturally, and both parties are contributing equally, it's a sign of mutual interest. If one person dominates the conversation or keeps steering it back to their own interests, it may indicate a lack of genuine interest.
5. Time spent: If the conversation lasts longer than expected or planned, it's a sign that both parties are engaged and interested.
6. Follow-up actions: If someone shows genuine interest, they may follow up with you after the conversation, whether it's through a message, email, or phone call.
Remember, however, that sometimes people may be polite or professional, even if they're not genuinely interested. It's essential to consider the context and the relationship you have with the person to determine their level of interest accurately. |

**(b) DP3O**

| | |
|---|---|
| Input → | Q: How can you determine if a person is genuinely interested in a conversation or simply being polite? |

| | |
|---|---|
| Mistral DP3O → | Here are some cues to help determine whether a person is genuinely interested in a conversation or simply being polite:
1. Active Listening: If someone is actively engaged in the conversation, asking questions, and responding thoughtfully, they are likely interested in the topic.
2. Body Language: Notice if the person maintains eye contact, leans in, or nods during the conversation. These nonverbal cues indicate engagement and interest.
3. Follow-up Questions: If the person asks follow-up questions to clarify or learn more, it suggests they are invested in the conversation.
4. Participation: Observe how involved the person is in the conversation. Are they sharing their own thoughts and experiences, or are they mostly offering short, polite responses?
5. Delay in Responding: If someone takes a pause to consider their response or seems genuinely interested in what the other person has to say, they may be more engaged than polite.
6. Topic Choice: If the person seems genuinely interested in the topic, they may bring up related topics or ask for more information to continue the conversation.
Remember, while these cues can help you determine if someone is genuinely interested, sometimes individuals may be polite but disengaged due to various reasons. Be mindful and open to the possibility that someone might be struggling to engage, and try to create an inclusive, welcoming environment. |

## C   Qualitative Study

Here we present two qualitative examples from AlpacaEval 2 generated by the Mistral model after training with DPO (Figure 9) and DP3O (Figure 9). These examples highlight the differences in response style and presentation quality between the two methods. Compared to the response produced by the DPO-trained model, the generation from DP3O is more structured and demonstrates a clearer hierarchical organization of information. Such organization improves the logical flow of the response and makes individual points easier to identify. As a result, the DP3O output is more readable and presents the content in a clearer and more accessible way.

