# OpenReview forum: "Towards Bridging the Gap Between Offline and Iterative Alignment via Preference Distillation"
_TMLR — Under review for TMLR_

### Review · Reviewer_PS83 · 2026-04-12

**Summary Of Contributions:**

- This work identifies explicit preference modeling itself as the key factor behind the success of iterative DPO approaches. Building on this insight, the proposed DP3O algorithm distills preference probabilities from an explicit teacher model.

- At a high level, the proposed DP3O algorithm feels quite similar to d-DPO; the relationship between the two is almost reminiscent of looking at the difference between standard knowledge distillation (mixing hard and soft targets) and logit distillation in classification tasks.

**Audience:**

Yes

**Audience Explanation:**

Yes, there is certainly a broad audience for this work; it addresses LLM preference alignment, which is currently a major and highly active research area.

**Broader Impact Concerns:**

I recommend adding a standard discussion on potential societal impacts, as is commonly expected for studies in the domain of LLM alignment.

**Claims And Evidence:**

No

**Claims Explanation:**

Currently, my response is 'No' due to reservations regarding the theoretical analysis. Detailed feedback on this matter is provided in the 'Requested Changes' section below; I am willing to listen to the authors' thoughts on this.

- The controlled experiments in Section 4.1 well demonstrate that Iterative DPO outperforms standard DPO purely due to the use of explicit preference modeling. If I had to point out a minor flaw, while using identical optimization settings is standard practice for a fair comparison, a more rigorous design would be to evaluate each method using its own optimal hyperparameters. However, that is a minor issue that does not undermine the core findings.

- The experiments are well-structured, incorporating essential baselines and demonstrating that the proposed DP3O, alongside d-DPO, occupies a strategic middle ground between standard DPO and iterative DPO in terms of the performance-cost trade-off. While some may favor even larger-scale evaluations, the results provided in the 7-8B parameter regime ensure that the study meets a credible threshold for modern LLM research.

- Overall, the empirical claims are solid. However, I believe a technical discussion with the authors regarding the theoretical analysis in Section 4.3 is a prerequisite before I can fully endorse the evidence as "convincing and clear." Specifically, certain assumptions in the convergence and bias-variance analysis require further justification to bridge the gap between the theoretical bounds and the empirical results.

**Requested Changes:**

- (Section 2) The manuscript asserts that d-DPO (Fisch et al., 2024) does 'not identify explicit preference modeling itself as the key advantage over implicit formulations.' However, Fisch et al. (2024) state that 'explicit reward modeling can, in fact, offer substantial practical and theoretical benefits.' I recommend toning down the claim; for instance, the authors could acknowledge that 'while prior work recognized the benefits of explicit reward modeling, ...'

- (Theorem 4.1) The theoretical analysis in Theorem 4.1 relies on the assumption of a finite hypothesis class $\mathcal{F}$ to bound the estimation error. However, since both the policy and reward models are parameterized by deep neural networks with billions of continuous parameters (e.g., Pythia-2.8B), this assumption is arguably non-representative of the actual learning dynamics; the resulting bound is highly vacuous. It would be beneficial for the authors to explicitly discuss this limitation or clarify that the theorem is intended to provide intuition rather than a rigorous guarantee.

- (Theorem 4.1) Moreover, the authors characterize the explicit class $\mathcal{P}$ as being 'smaller' than the implicit class $\Pi$ based on the boundedness of output logits. However, given that both parameterizations leverage the same large-scale backbone, their functional capacity and intrinsic capacity remain virtually identical. It is questionable whether the mere boundedness of output logits is sufficient to claim a significantly 'richer' hypothesis class for the implicit formulation in a way that impacts the bound.

- (Theorem 4.1) As far as I know, to adapt the theoretical results of Zhang (2006), an explicit complexity regularization is required to guarantee the convergence bounds. However, the current statement apperas to be a standard MLE without such explicit regularization term; the authors should justify whether Zhang's theory is directly applicable to the presented setup.

- (Theorem 4.4) I suggest clarifying that the bound is a shared across DPO and DP3O; reminding that DPO is a special case of this bound (when $\alpha=1$) would provide a more intuitive bridge for the reader.

- (Theorem 4.4) Even if we set aside the aforementioned issues, the claim that 'DP3O can generalize better than DPO' remains controversial. The bound in Theorem 4.4, which corresponds to the Equation 9 of Menon et al. (2021), shows that while DP3O reduces the second term (variance), it necessarily introduces the third term (bias) that vanishes in the DPO case. Currently, we don't have any evidence that the variance reduction outweighs the introduced bias.

---

> ### Author Response · Authors · 2026-05-29
>
> We are very grateful for your valuable comments and insightful feedback! In response to your comments, we make the following clarifications and additional revisions to the paper. Below, we list your comments in quotes and provide our point-by-point responses. Please refer to the latest submission for the revised paper, taking all your suggestions using **blue** color.
>
> ***Responses to Requested Changes***
>
> >1. (Section 2) The manuscript asserts that d-DPO (Fisch et al., 2024) does 'not identify explicit preference modeling itself as the key advantage over implicit formulations.' However, Fisch et al. (2024) state that 'explicit reward modeling can, in fact, offer substantial practical and theoretical benefits.' I recommend toning down the claim; for instance, the authors could acknowledge that 'while prior work recognized the benefits of explicit reward modeling, ...'
>
> Thank you for your suggestion. We have revised the RLHF Algorithms paragraph in **Section 2** as follows:
>
> "While d-DPO recognizes the benefits of explicit reward modeling, our controlled experiments further identify it as a substantial factor in the offline–iterative gap, and we show both empirically and theoretically that distilling preference probabilities offers advantages over hard labels."
>
> >2. (Theorem 4.1) The theoretical analysis in Theorem 4.1 relies on the assumption of a finite hypothesis class to bound the estimation error. However, since both the policy and reward models are parameterized by deep neural networks with billions of continuous parameters (e.g., Pythia-2.8B), this assumption is arguably non-representative of the actual learning dynamics; the resulting bound is highly vacuous. It would be beneficial for the authors to explicitly discuss this limitation or clarify that the theorem is intended to provide intuition rather than a rigorous guarantee.
>
> Thank you for this important critique. We agree that the finite-class assumption is a simplification of real LLMs architectures, and we have revised **Section 4.3** to incorporate limitations:
>
> “Since real preference models are usually large neural networks with high-dimensional parameter spaces rather than a literal finite class, Theorem 4.1 mainly captures the scaling of estimation error with the effective hypothesis-class size, rather than providing a sharp finite-sample guarantee.”
>
>
> >3. (Theorem 4.1) Moreover, the authors characterize the explicit class as being 'smaller' than the implicit class based on the boundedness of output logits. However, given that both parameterizations leverage the same large-scale backbone, their functional capacity and intrinsic capacity remain virtually identical. It is questionable whether the mere boundedness of output logits is sufficient to claim a significantly 'richer' hypothesis class for the implicit formulation in a way that impacts the bound.
>
> Thank you for this important comment. We agree that the boundedness of output logits alone is not sufficient to justify the claim that the implicit class is significantly richer in a way that affects the bound. We have therefore removed this argument from the main text. Since analytically comparing two deep model classes with the same trainable backbone is difficult, we now support the discussion after Theorem 4.1 with two more concrete pieces of evidence.
>
> First, we compare the output-level parameterizations of the two preference models. Since the backbone is the same and trainable in both cases, our comparison is not about the total number of trainable parameters, but about two differences at the output level.
>
> **1. Different last-layer parameterizations.** The explicit model uses a scalar reward head $\mathbb{R}^d \to \mathbb{R}$, while the implicit preference model induced by DPO uses the language-modeling head $\mathbb{R}^d \to \mathbb{R}^V$. For Pythia-1.4B, where $d=2048$ and $V=50304$, the explicit reward head has roughly 2K parameters, whereas the LM head used by the implicit parameterization has roughly 103M parameters.
>
> **2. Different reward representations.** The explicit model assigns each response a scalar reward and computes preferences through the difference between two response-level rewards. In contrast, the implicit model defines the reward through a sum of per-token log-ratios between the policy and the reference model over the vocabulary. Therefore, the implicit parameterization represents preference information through a much higher-dimensional output structure. This suggests that the implicit parameterization has more flexibility for fitting preference labels.

---

> > ### Author Response · Authors · 2026-05-29
> >
> > Second, we add an empirical randomization test in **Appendix A.4**, following [1]. We randomly reassign the preference labels and train the explicit and implicit preference models with trainable backbones, using the same initialization, optimizer, regularization, and matched $\beta$ setting. In this experiment, the implicit model drives the training loss well below the random-guessing, while the explicit model remains close to random-guessing. This provides empirical evidence that, in our setup, the implicit parameterization is more prone to memorizing random preference labels.
> >
> > Taken together, the two pieces of evidence show that the implicit hypothesis class $\Pi$ can be strictly richer than the explicit class $\mathcal{P}$.
> >
> > We have added the structural comparison, the randomization-test protocol, and the corresponding learning curves to **Appendix A.4**. We have also revised the discussion following Theorem 4.1 in **Section 4.3** to replace the bounded-logit argument with this revised interpretation.
> >
> > [1] Zhang, Chiyuan, et al. "Understanding deep learning requires rethinking generalization." International Conference on Learning Representations, 2017.
> >
> >
> > >4. (Theorem 4.1) As far as I know, to adapt the theoretical results of Zhang (2006), an explicit complexity regularization is required to guarantee the convergence bounds. However, the current statement apperas to be a standard MLE without such explicit regularization term; the authors should justify whether Zhang's theory is directly applicable to the presented setup.
> >
> > We thank the reviewer for pointing this out. We agree that the general information-complexity framework of Zhang (2006) involves an explicit KL-entropy complexity regularization term and should not be directly interpreted as a guarantee for arbitrary unregularized MLEs over continuous or unrestricted model classes.
> >
> > Our proof, however, does not directly invoke the full theorem of Zhang (2006). Instead, it uses the similar likelihood-ratio/Hellinger-affinity idea in a finite-class preference modeling setting. In this setting, if we take a uniform prior over the finite class $\mathcal F$, then the KL-entropy penalty for any deterministic estimator selecting $p\in\mathcal F$ is $D_{\mathrm{KL}}(\delta_p\|\pi)=\log|\mathcal F|$, which is independent of $p$. Therefore, adding this penalty does not change the optimizer, and the resulting estimator coincides with the standard MLE used in Theorem 4.1.
> >
> > We have revised the **Appendix B.1** to clarify this finite-class reduction and its connection to Zhang (2006).
> >
> > >5. (Theorem 4.4) I suggest clarifying that the bound is a shared across DPO and DP3O; reminding that DPO is a special case of this bound (when) would provide a more intuitive bridge for the reader.
> >
> > We agree that this framing is much cleaner and we have rewritten **Theorem 4.4** accordingly:
> > “Note that this bound is shared by DPO and DP3O: DPO corresponds to the special case $\alpha = 1$, while DP3O is the general $\alpha \in [0, 1)$ case.”
> >
> > >6. (Theorem 4.4) Even if we set aside the aforementioned issues, the claim that 'DP3O can generalize better than DPO' remains controversial. The bound in Theorem 4.4, which corresponds to the Equation 9 of Menon et al. (2021), shows that while DP3O reduces the second term (variance), it necessarily introduces the third term (bias) that vanishes in the DPO case. Currently, we don't have any evidence that the variance reduction outweighs the introduced bias.
> >
> > Thank you for this important critique. We agree that our theory does not guarantee that the variance reduction always outweighs the introduced bias. We have addressed this concern in two ways.
> >
> > First, we have softened the theoretical claim in the revised **Section 4.3.2**. We no longer present DP3O's better generalization over DPO as a theorem-level conclusion. Instead, we present Theorem 4.4 as a bias-variance tradeoff: DP3O may reduce the variance term relative to DPO, as suggested by Lemma 4.3, but it also introduces a teacher-discrepancy term that vanishes for DPO. Whether this tradeoff is favorable depends on the teacher quality and the data distribution. In our experiments, the explicit preference model appears sufficiently accurate, and the improvement of DP3O over DPO is consistent with this favorable regime.
> >
> > Second, the $\alpha$-ablation provides a direct empirical test of this tradeoff. A natural interpretation of the bound is to ask whether the variance reduction outweighs the bias cost in practice. In the $\alpha$-ablation study in Figure 3, all tested $\alpha<1$ settings outperform $\alpha=1$ at the final evaluation checkpoint, where $\alpha=1$ corresponds to standard DPO. This provides empirical support that, in the regimes we evaluate, the reduction in variance can outweigh the additional bias.
> >
> > [1] Zhang, Chiyuan, et al. "Understanding deep learning requires rethinking generalization." International Conference on Learning Representations. 2017.

---

### Review · Reviewer_o37D · 2026-04-15

**Summary Of Contributions:**

The paper studies the problem of preference optimization. Particularly, the paper tries to identify why it is the case that the iterative preference optimization typically works better than the offline DPO. The paper claims that the major gap is not a result of online v.s. offline but more from the use of explicit v.s. implicit preference model. The paper studies in controlled setting that if a golden-label explicit preference model is given, even with offline training, DP3O can get quite improved performance. The paper then extends the setting to first train an explicit preference model and to larger models/datasets and show that DP3O improves upon DPO.

**Audience:**

Yes

**Audience Explanation:**

The paper is about preference optimization and DPO, which are important and practical RL topics, and therefore should be of interests to the committee.

**Claims And Evidence:**

Yes

**Claims Explanation:**

Strengths:
1) The paper studies an important and practical problem. The paper also proposes an interesting perspective to compare DPO and iterative methods.
2) The paper proposes a simple method DP3O that has improved performance compared to DPO.
3) The paper also provides theoretical justifications for their approach.


Weaknesses:
1) The proposed method DP3O introduces an additional explicit preference model. In some sense, this contradicts with the purpose of DPO, where we want to directly learn from the preference data. This also introduces the caveat: the proposed method greatly relies on the quality of the explicit preference model.

2) I am not fully convinced that the major gap between iterative methods and DPO come from the explicit preference model. I feel that the major gain of D3PO is to utilize the soft labels provided in the explicit preference model. However, that soft label is not used in the iterative methods. Two more experiments might be valuable: 1) how does iterative methods behave without their explicit preference model and 2) how does iterative methods behave utilizing the soft labels of the explicit preference model?

3) DP3O has improved upon DPO, but in some cases the performance is still lower than the iterative methods.

**Requested Changes:**

Please address the weaknesses above.

---

> ### Author Response · Authors · 2026-05-29
>
> We are very grateful for your valuable comments and insightful feedback! In response to your comments, we make the following clarifications and additional revisions to the paper. Below, we list your comments in quotes and provide our point-by-point responses. Please refer to the latest submission for the revised paper, taking all your suggestions using **blue** color.
>
> ***Responses to Weakness***
>
> >1. The proposed method DP3O introduces an additional explicit preference model. In some sense, this contradicts with the purpose of DPO, where we want to directly learn from the preference data. This also introduces the caveat: the proposed method greatly relies on the quality of the explicit preference model.
>
>
> Thank you for raising this thoughtful concern. We agree that DP3O introduces an additional explicit preference model, and therefore is not as minimal as standard DPO. However, our goal is to understand why iterative DPO often performs better than offline DPO, and to bring part of that benefit back to the offline setting. As discussed in Section 4.1, iterative DPO commonly uses an explicit preference model to score or relabel newly generated responses [1,2]. Therefore, DP3O does not introduce an unrelated extra component; instead, it uses the same type of preference information that is already used in iterative methods, but distills it directly into the offline DPO objective.
>
> This design avoids the repeated response generation and re-annotation steps required by iterative methods. As shown in Table 1, DP3O improves over standard offline DPO and reduces training time by approximately **42%** compared with final-iteration iterative DPO. Thus, DP3O is intended as a more efficient middle ground between standard offline DPO and iterative DPO, rather than as a method that removes all extra components.
>
> Regarding the dependence on the quality of the explicit preference model, we agree that this is an important issue. To study it, we added a misspecification experiment in **Appendix A.3**. In this experiment, we use an LLM-as-judge as the ground-truth preference model, which does not necessarily follow the Bradley-Terry form used by our explicit preference model. Even in this setting, the explicit preference model achieves better accuracy and calibration, and DP3O achieves the highest downstream win rate.
>
> These results suggest that **DP3O does not require the explicit preference model to be perfectly specified**. Instead, it can still benefit from the preference probabilities provided by the explicit model, while keeping the training procedure much cheaper than iterative alignment.
>
>
>
> >2. I am not fully convinced that the major gap between iterative methods and DPO comes from the explicit preference model. I feel that the major gain of D3PO is to utilize the soft labels provided in the explicit preference model. However, that soft label is not used in the iterative methods. Two more experiments might be valuable: 1) how does iterative methods behave without their explicit preference model and 2) how does iterative methods behave utilizing the soft labels of the explicit preference model?
>
> Thank you for this insightful comment. We agree that the effects of explicit preference modeling and soft-label distillation should be separated more carefully. To address this, we added a new ablation study in **Appendix A.5**. The study compares three iterative variants under the same prompt split, generated candidate responses, number of iterations, and optimization hyperparameters: iterative DPO with an explicit preference model, iterative DPO with an implicit preference model, and iterative DP3O with soft labels from the explicit preference model.
>
> The new results show two separate effects.
>
> **1. Explicit preference model improves the response selection step in iterative DPO.** When we replace the explicit preference model with the implicit preference model induced by the policy, the final-iteration win rate drops from $72.69\%$ to $66.76\%$. This suggests that the explicit preference model is an important contributor to the offline-iterative gap, beyond the effect of iterative data generation alone.
>
> **2. Soft preference probabilities provide an additional benefit on top of hard relabeling.** Under the same explicit preference model, iterative DP3O uses the soft preference probabilities, while iterative DPO converts the preference signal into hard winner--loser labels. Iterative DP3O achieves a final-iteration win rate of $74.25\%$, compared with $72.69\%$ for iterative DPO. This shows that soft-label distillation contains useful information beyond the hard labels used by standard iterative DPO.

---

> > ### Author Response · Authors · 2026-05-29
> >
> > We also include complementary evidence from the offline teacher-type ablation in **Figure 3 (right)**. In this experiment, both variants use the same offline data and the same soft-label objective; the only difference is whether the teacher is explicit or implicit. DP3O with the explicit teacher improves the average win rate from $64.99\%$ to $67.95\%$ and the reward gain from $1.28$ to $1.39$ compared with DP3O using the implicit teacher. **This suggests that the improvement is not only due to using soft labels, but also depends on the parameterization of the teacher model.**
> >
> > Overall, these results clarify the two factors raised by the reviewer. Iterative DPO benefits from using an explicit preference model rather than an implicit one for response selection and hard relabeling, and using the explicit model's soft probabilities can further improve performance. We also clarify in **Section 4.1** that explicit preference modeling is not the only reason for the offline-iterative gap: on-policy data generation is also important, as shown in prior work. Our claim is that explicit preference modeling is another underexplored contributor.
> >
> > We further support this claim in **Appendix A.4** by comparing the explicit and implicit preference classes structurally and empirically. The results suggest that the implicit parameterization is more flexible for fitting preference labels and more prone to memorization, which can lead to larger estimation error.
> >
> > >3. DP3O has improved upon DPO, but in some cases the performance is still lower than the iterative methods.
> >
> > We agree that DP3O does not always outperform iterative methods in terms of task performance. Our intention is not to position DP3O as a complete replacement for iterative alignment, but rather as a middle ground between offline DPO and iterative DPO. Compared with standard DPO, DP3O benefits from explicit preference modeling and achieves stronger performance, while avoiding the repeated response generation and annotation required by iterative methods. Therefore, DP3O provides a more efficient alternative that narrows the gap between offline and iterative alignment, even though iterative methods may still achieve higher performance in some settings.
> >
> > We have rewritten the discussion at the **end of Section 5.2** to make this trade-off explicit.
> >
> >
> > [1] Xiong, Wei, et al. "Iterative Preference Learning from Human Feedback: Bridging Theory and Practice for RLHF under KL-constraint." International Conference on Machine Learning. PMLR, 2024.
> >
> > [2] Pang, Richard Y., et al. "Iterative reasoning preference optimization." Advances in Neural Information Processing Systems 37 (2024): 116617-116637.

---

### Review · Reviewer_bPyJ · 2026-07-05

**Summary Of Contributions:**

Paper studies why iterative DPO often outperforms offline DPO and argues that a key reason is the use of an explicit preference model. Based on this insight, the authors propose DP3O, an offline alignment method that first trains an explicit preference model and then distills its soft preference probabilities into the DPO objective. The paper provides controlled experiments, theory, and empirical comparisons showing that DP3O improves over offline baselines.

**Audience:**

Yes

**Audience Explanation:**

RL Alignment is of interest to the broader community.

**Claims And Evidence:**

Yes

**Claims Explanation:**

Strength:

The proposed method is simple, easy to implement, and naturally connects DPO with preference-model distillation. I also appreciate the controlled experiments isolating explicit preference modeling, as they make the paper more convincing than a purely benchmark-driven contribution. The comparison with DPO, CDPO, RDPO, d-DPO, and iterative DPO is useful, and the theory gives a reasonable bias–variance interpretation.

Weakness:

The main empirical claim in paper, depends on the quality of the explicit preference model. The paper would be stronger if it more clearly characterized failure cases: for example, when the preference model is miscalibrated, smaller than the policy, trained on noisy labels, or distribution-shifted from the policy outputs. Also, the controlled setup uses a golden reward model for annotation/evaluation, which is clean but may partially favor methods that better mimic the reward model's probabilities. Finally, the efficiency comparison should clearly include the cost of training the explicit preference model, not only policy optimization time.

Questions:

1) How robust is DP3O when the explicit preference model is poorly calibrated or less accurate than the implicit DPO preference model on some regions of the data?

2) Does the optimal value of the interpolation parameter α correlate with preference-model accuracy? Could α be chosen adaptively per sample using uncertainty or calibration?

3) Since DP3O relies on soft probabilities from the explicit model, how does it behave under noisy or ambiguous human preference labels where the “teacher” may become overconfident?

**Requested Changes:**

Please see above.

---

> ### Author Response · Authors · 2026-07-18
>
> We are very grateful for your valuable comments and insightful feedback! In response to your comments, we made the following clarifications and additional revisions to the paper. Below, we summarize your comments in quotes and provide our point-by-point responses. Please refer to the latest submission for the revised paper, taking all your suggestions using **blue** color.
>
> ***Responses to Weaknesses***
>
> > Characterize failure cases: preference model miscalibrated, trained on noisy labels, or distribution-shifted.
>
> Thanks for raising these points. We have in fact considered these settings, and restate them here to make the picture clear:
> - **Noisy labels (Table 2).** We inject label-flipping noise at $\epsilon \in \{0.3, 0.45\}$, which directly corrupts the teacher. DP3O is the **most robust** method: hard-label iterative DPO relies on the teacher's argmax labels and degrades sharply, whereas DP3O's soft probabilities still carry useful uncertainty and it retains the best win rate and reward gap.
> - **Miscalibration / misspecification (Appendix A.3, Table 5).** Here the true preference function is a *non-BT* LLM-as-judge, so the BT teacher is misspecified and distribution-shifted from the true labels. We report calibration via **ECE**: the explicit teacher still attains the best accuracy/calibration trade-off (ACC 82.4, **ECE 0.24**) and the highest downstream win rate, outperforming both prompt-based and trained LLM-as-judge alternatives.
>
> We would also emphasize that DP3O is designed as an *interpolation* method: through $\alpha$ it blends the teacher's soft probabilities with the ground-truth hard labels, rather than committing to the teacher alone. This is a built-in safeguard, a severely degraded teacher can be down-weighted by raising $\alpha$, and at $\alpha=1$ its influence is cancelled entirely to recover standard DPO. Theorem 4.4 makes this precise: the teacher-error term scales with $(1-\alpha)$ and vanishes as $\alpha \to 1$.
>
> > The golden reward model may partially favor methods that better mimic its probabilities.
>
> In controlled experiments, the gold model not as an artifact to be avoided but as the ground-truth signal every method is trying to recover. Once a true reward (or true preference) is fixed as the target, better recovering the true probabilities is precisely what it *means* to be a better method — just as in RL, where the method attaining higher task performance under the true reward is, by definition, the better method for that domain. The substantive question is therefore not *whether* a method aligns with the true signal, but *how well* it recovers it, especially when the learning signal is imperfect. This is what our experiments show: DP3O recovers the true signal more faithfully than the baselines across three regimes of increasing difficulty: (1) **clean labels** (main controlled experiments), (2) **noisy labels** (Table 2, $\epsilon \in \{0.3, 0.45\}$), and (3) **misspecification** (Appendix A.3, non-BT reward model), attaining the best downstream performance in every case.
>
> > Efficiency should include the cost of training the explicit preference model.
>
> We would clarify that we have already accounted for this. Training the explicit preference model takes only a single epoch, and the wall-clock times in Table 1 **already include this cost for both DP3O and iterative DPO** (iterative DPO also relies on an explicit model to relabel its on-policy responses). Even with the explicit-model training folded into both, DP3O (1.55 h) is still far more efficient than iterative DPO (2.67 h), since it avoids the repeated rollout-and-relabel passes that dominate iterative DPO's cost.
>
> ***Responses to Questions***
>
> > Q1. Robustness when the teacher is poorly calibrated or less accurate than the implicit model on some regions?
>
> Thank you for raising this robustness question. If the explicit preference model is poorly calibrated or locally less accurate than the implicit DPO model, DP3O may inherit biased supervision in those regions and is not guaranteed to outperform DPO. However, DP3O interpolates the teacher's soft probabilities with the original hard labels through $\alpha$; increasing $\alpha$ reduces reliance on the teacher, and $\alpha=1$ exactly recovers DPO. This corresponds to the bias–variance trade-off in our theory: soft targets reduce variance, while teacher error introduces a discrepancy term $(1-\alpha)\|\hat p - p^\star\|_2$ weighted by $1-\alpha$.
>
> Importantly, **DP3O does not assume that an arbitrary explicit teacher is always better**. Our core motivation is that, with the same data and backbone, the explicit parameterization admits better estimation-error control and empirically produces more accurate preference probabilities than the implicit DPO model, providing a simple way to construct a stronger teacher on average.

---

> > ### Author Response · Authors · 2026-07-18
> >
> > > Q2. Does optimal α correlate with teacher accuracy? Could α be adaptive per sample?
> >
> > A more accurate teacher should generally make stronger distillation safer, although Theorem 4.4 does not establish a strictly monotonic relationship between teacher accuracy and the optimal $\alpha$. The bound shows that the teacher-discrepancy penalty scales with $(1-\alpha)\|\hat p - p^\star\|_2$. Therefore, when the teacher error is smaller, assigning more weight to its soft probabilities, that is, using a smaller $\alpha$, incurs a smaller bias penalty. However, Theorem 4.4 alone does not determine the optimal $\alpha$, since the empirical-risk and variance terms may also vary with $\alpha$. Figure 3 demonstrates that performance is sensitive to $\alpha$, while Table 2 shows that DP3O remains robust as the teacher degrades under label noise; neither experiment directly measures the correlation between teacher accuracy and the optimal $\alpha$.
> >
> > A sample-adaptive $\alpha_i$ is a natural extension. Ideally, the method would **place less weight on the teacher for samples on which it is unreliable**. In practice, held-out calibration, predictive entropy, or disagreement between the explicit and implicit preference models could serve as uncertainty proxies. We have added this as a future direction in conclusion section of revised paper.
> >
> > > Q3. Behavior under noisy/ambiguous labels where the teacher becomes overconfident?
> >
> > An overconfident teacher can be harmful, especially on ambiguous samples where the true preference probability is close to $0.5$. If the teacher incorrectly assigns a probability close to $0$ or $1$, its soft target introduces a larger teacher-discrepancy bias and no longer preserves the uncertainty information that motivates distillation. DP3O reduces its reliance on such a teacher by interpolating the teacher probability with the observed hard label through $\alpha$: a larger $\alpha$ places more weight on the original label, and $\alpha=1$ recovers standard DPO. This behavior is consistent with the teacher-discrepancy term in Theorem 4.4.
> >
> > Empirically, DP3O remains the best-performing offline method under label-flipping ratios of $0.3$ and $0.45$. These results indicate that soft preference probabilities can still provide useful supervision under substantial label noise, while the interpolation with hard labels offers additional robustness when the teacher becomes less reliable.